# Neural attentional filters and behavioural outcome follow independent individual trajectories over the adult lifespan

Sarah Tune[1,2]*, Jonas Obleser[1,2]*

[1]Center of Brain, Behavior, and Metabolism, University of Lübeck, Lübeck, Germany; [2]Department of Psychology, University of Lübeck, Lübeck, Germany

**Abstract** Preserved communication abilities promote healthy ageing. To this end, the age-typical loss of sensory acuity might in part be compensated for by an individual's preserved attentional neural filtering. Is such a compensatory brain–behaviour link longitudinally stable? Can it predict individual change in listening behaviour? We here show that individual listening behaviour and neural filtering ability follow largely independent developmental trajectories modelling electroencephalographic and behavioural data of $N$ = 105 ageing individuals (39–82 y). First, despite the expected decline in hearing-threshold-derived sensory acuity, listening-task performance proved stable over 2 y. Second, neural filtering and behaviour were correlated only within each separate measurement timepoint (T1, T2). Longitudinally, however, our results raise caution on attention-guided neural filtering metrics as predictors of individual trajectories in listening behaviour: neither neural filtering at T1 nor its 2-year change could predict individual 2-year behavioural change, under a combination of modelling strategies.

*For correspondence:
sarah.tune@uni-luebeck.de (ST);
jonas.obleser@uni-luebeck.de
(JO)

## eLife assessment

This study provides a **valuable** contribution to understanding the neural mechanisms underlying age-related changes in attention and speech understanding. The large dataset ($N$ = 105) provides **convincing** evidence for how speech recognition behaviour and neural tracking of speech separately evolve in about 2 y. The work would be of interest to psychologists, neuroscientists, and audiologists.

## Introduction

Speech comprehension is an essential aspect of human communication, enabling us to understand and interact with others effectively. Preserved communication is therefore critical to social well-being and healthy ageing (*Lindenberger, 2014*). Any translational advance aimed at maintaining and restoring successful cognitive ageing crucially relies on understanding the factors that explain and predict individual trajectories of listening performance. However, the evidence on these potential factors is astonishingly limited.

As we age, our ability to comprehend speech can decline due to age-related changes in the auditory system (i.e. sensory acuity) and in cognitive resources. Age-related hearing loss reduces the ability to detect and discriminate speech sounds, especially in noisy environments. However, it has long been recognised that increasing age and hearing loss cannot fully account for the considerable degree of inter-individual differences observed in listening behaviour and its lifespan change (*Akeroyd, 2008*; *Houtgast and Festen, 2008*; *Humes et al., 2012*).

**eLife digest** Humans are social animals. Communicating with other humans is vital for our social wellbeing, and having strong connections with others has been associated with healthier aging. For most humans, speech is an integral part of communication, but speech comprehension can be challenging in everyday social settings: imagine trying to follow a conversation in a crowded restaurant or decipher an announcement in a busy train station. Noisy environments are particularly difficult to navigate for older individuals, since age-related hearing loss can impact the ability to detect and distinguish speech sounds. Some aging individuals cope better than others with this problem, but the reason why, and how listening success can change over a lifetime, is poorly understood.

One of the mechanisms involved in the segregation of speech from other sounds depends on the brain applying a 'neural filter' to auditory signals. The brain does this by aligning the activity of neurons in a part of the brain that deals with sounds, the auditory cortex, with fluctuations in the speech signal of interest. This neural 'speech tracking' can help the brain better encode the speech signals that a person is listening to.

Tune and Obleser wanted to know whether the accuracy with which individuals can implement this filtering strategy represents a marker of listening success. Further, the researchers wanted to answer whether differences in the strength of the neural filtering observed between aging listeners could predict how their listening ability would develop, and determine whether these neural changes were connected with changes in people's behaviours.

To answer these questions, Tune and Obleser used data collected from a group of healthy middle-aged and older listeners twice, two years apart. They then built mathematical models using these data to investigate how differences between individuals in the brain and in behaviours relate to each other. The researchers found that, across both timepoints, individuals with stronger neural filtering were better at distinguishing speech and listening. However, neural filtering strength measured at the first timepoint was not a good predictor of how well individuals would be able to listen two years later. Indeed, changes at the brain and the behavioural level occurred independently of each other.

Tune and Obleser's findings will be relevant to neuroscientists, as well as to psychologists and audiologists whose goal is to understand differences between individuals in terms of listening success. The results suggest that neural filtering guided by attention to speech is an important readout of an individual's attention state. However, the results also caution against explaining listening performance based solely on neural factors, given that listening behaviours and neural filtering follow independent trajectories.

Recent research has focused on the neurobiological mechanisms that promote successful speech comprehension by implementing 'neural filters' that segregate behaviourally relevant from irrelevant sounds. Such neural filter mechanisms act by selectively increasing the sensory gain for behaviourally relevant inputs or by inhibiting the processing of irrelevant inputs (*Cherry, 1953*; *Broadbent, 1958*; *Wöstmann et al., 2020*). A growing body of evidence suggests that speech comprehension is neurally supported by an attention-guided filter mechanism that modulates sensory gain and arises from primary auditory and perisylvian brain regions: by synchronising its neural activity with the temporal structure of the speech signal of interest, the brain 'tracks' and thereby better encodes behaviourally relevant auditory inputs to enable attentive listening (*Ding and Simon, 2012*; *Zion Golumbic et al., 2013*; *Obleser and Kayser, 2019*; *Lakatos et al., 2008*).

In a large age-varying cohort ($N$ = 155; 39–80 y), we have previously shown how the fidelity of this neural filtering strategy can help explain differences in listening behaviour (i) from individual to individual and (ii) within individuals from sentence to sentence (*Tune et al., 2021*; *O'Sullivan et al., 2015*; *Mesgarani and Chang, 2012*). As participants performed a challenging dual-talker listening task, we recorded their electroencephalogram (EEG). We observed that enhanced neural filtering—defined as stronger neural tracking of attended vs. ignored speech—led to more accurate and overall faster responses. Notably, we observed both neural filtering as well as its link to behaviour to be independent of chronological age and severity of hearing loss (*Tune et al., 2021*; *Figure 1A*).

The observation of such brain–behaviour relationships critically advances our understanding of the neurobiological foundation of cognitive functioning by showing, for example, how neural

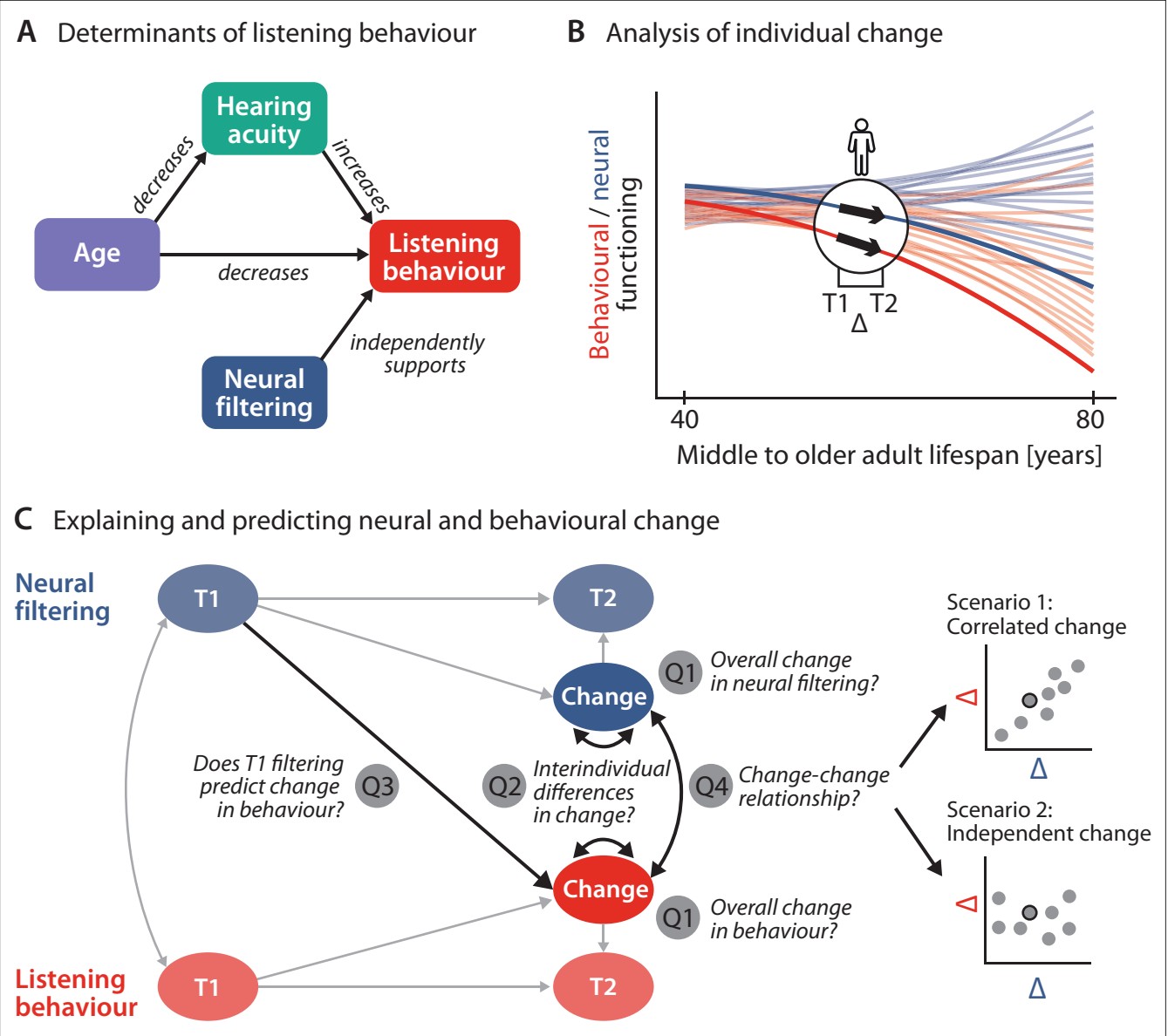

**Figure 1.** Schematic illustration of key assumptions and research questions. (**A**) Listening behaviour at a given timepoint is shaped by an individual's sensory and neural functioning. Increased age decreases listening behaviour both directly and indirectly via age-related hearing loss. Listening behaviour is supported by better neural filtering ability, independently of age and hearing acuity. (**B**) Conceptual depiction of individual 2-year changes along the neural (blue) and behavioural (red) domain. Thin coloured lines show individual trajectories across the adult lifespan, while thick lines and black arrows highlight 2-year changes in a single individual. (**C**) Left schematic diagram highlighting the key research questions detailed in the introduction and how they are addressed in the current study using latent change score modelling. Right: across individuals, co-occurring changes in the neural and behavioural domain may be correlated (top) or independent of one another (bottom).

implementations of auditory selective attention support attentive listening. They also provide fruitful ground for scientific inquiries into the translational potential of neural markers. However, the potency of neural markers to predict future behavioural outcomes is often only implicitly assumed but seldom put to the test (*Woo et al., 2017*).

Using auditory cognition as a model system, we here aim to overcome this limitation by testing directly the hitherto unknown longitudinal stability of neural filtering as a neural compensatory mechanism upholding communication success. Going further, we ask to what extent an individual's attentional neural-filtering ability measured at a given moment is able to predict their future trajectory in listening performance. Only if this is the case, and only if such an association can plausibly be assumed

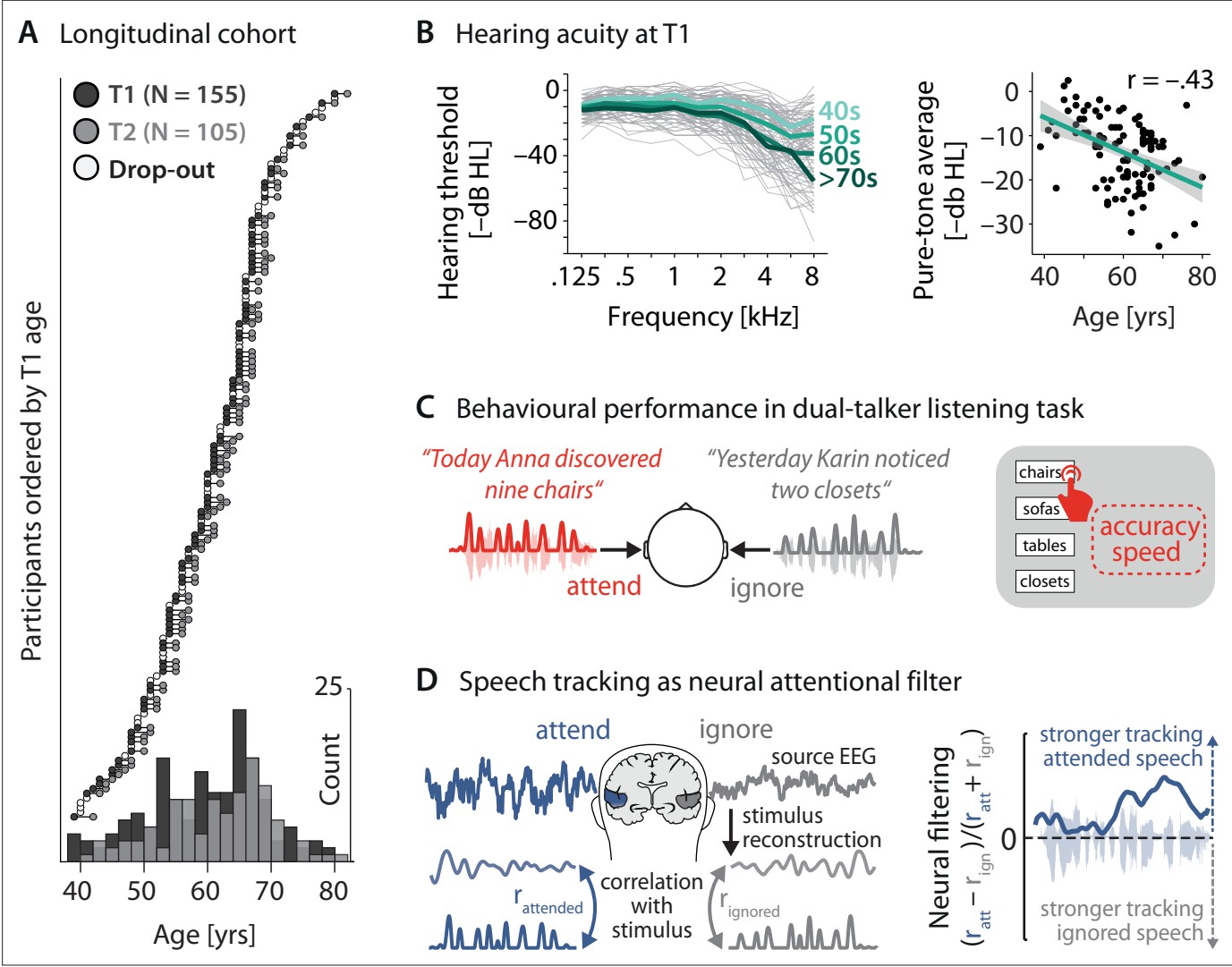

**Figure 2.** Key neural and behavioural metrics derived from a longitudinal cohort. (**A**) Longitudinal cohort of healthy middle-aged and older adults measured twice, 2 y apart. Circles represent individual participants at a given measurement time (dark grey: timepoint [T] 1, light grey: T2, white: dropouts after T1). Bottom: age distribution at T1 and T2 across 5-year bins. (**B**) Left: T1($N$ = 155) air conduction hearing thresholds per individual (thin grey lines) and age group (thick coloured lines). Note that for didactic purposes, throughout the article, thresholds are expressed as –dB HL to highlight the decrease in hearing acuity with age (left). Right: Pure-tone average hearing acuity (0.5, 1, 2, and 4 kHz across both ears; higher is better) negatively correlates with age ($N$ = 155; $r$ = –0.43, p=3.73 × 10$^{-6}$). (**C**) Participants listened to two sentences presented simultaneously to the left and right ear. In 50% of trials, a preceding visual cue indicated the to-be-attended target sentence. Listening behaviour is quantified via the accuracy and speed in identifying the final word of the target sentence. (**D**) Left: neural speech tracking as a proxy of an individual's neural filtering ability. Stimulus envelopes of attended and ignored sentences were reconstructed from source-localised electroencephalogram (EEG) activity in auditory cortex (see 'Materials and methods' for details) and correlated with the actual envelopes. Right: better neural filtering results from stronger neural tracking of attended compared to ignored speech. We analysed neural filtering derived from the entire sentence presentation period.

The online version of this article includes the following figure supplement(s) for figure 2:

**Figure supplement 1.** Experimental design and procedure.

to be causal for future changes in communication ability, neural filtering would be a potential translational target.

We here aim to fill this gap by analysing 2-year changes in the sensory, neural, and behavioural domain in a longitudinal subsample ($N$ = 105; 39–82 y) of the original AUDADAPT cohort (***Tune et al., 2021***; ***Figure 2A***). We apply a combination of advanced cross-sectional and longitudinal modelling strategies to address the following specific research questions (***Figure 1***).

First, by focusing on each domain individually, we ask how sensory, neural, and behavioural functioning evolve cross-sectionally across the middle and older adult lifespan (*Figure 1B*). More importantly, we also ask how they change longitudinally across the studied 2-year period (*Figure 1C*, Q1), and whether ageing individuals differ significantly in their degree of change (Q2). We expect individuals' hearing acuity and behaviour to decrease from T1 to T2. Since we previously observed inter-individual differences in neural filtering to be independent of age and hearing status, we did not expect any systematic longitudinal change in neural filtering.

Second, we test the longitudinal stability of the previously observed age- and hearing-loss-independent effect of neural filtering on both accuracy and response speed (*Figure 1A*). To this end, we analyse the multivariate direct and indirect relationships of hearing acuity, neural filtering, and listening behaviour within and across timepoints.

Third, leveraging the strengths of latent change score modelling (LCSM) (*McArdle, 2009*; *Kievit et al., 2018*), we fuse cross-sectional and longitudinal perspectives to probe the role of neural filtering as a precursor of behavioural change in two different ways: we ask whether an individual's T1 neural filtering strength can predict the observed behavioural longitudinal change (Q3), and whether 2-year change in neural filtering can explain concurrent change in listening behaviour (Q4). Here, irrespective of the observed magnitude and direction of T1–T2 developments, two scenarios are conceivable: Intra-individual neural and behavioural change may be either be correlated—lending support to a compensatory role of neural filtering—or instead follow independent trajectories (*Oschwald et al., 2019*; *Figure 1C*).

Answering these questions is vital for understanding the neurobiological mechanisms of successful communication across the lifespan. Answering them will also critically inform the development of interventions targeted at maintaining or restoring communication success and therefore concerns basic and applied researchers alike.

## Results

We studied an age-varying cohort of healthy middle-aged and older adults longitudinally (*N* = 105, 39–82 y, median age at T2: 63 y). The mean time difference between the two measurement timepoints reported here was 23.2 (SD: 4.0) mo. We characterise the multivariate relationship of key measures of sensory, neural, and behavioural functioning to explain and predict individual trajectories of listening performance.

At each of two measurement timepoints, participants underwent audiological assessment followed by EEG recording during which they performed a difficult dual-talker dichotic listening task (*Figure 2A and B*; *Tune et al., 2021*; *Alavash et al., 2021*; *Alavash et al., 2019*). In each trial of the task, participants listened to two temporally aligned but spatially separated five-word sentences. They then had to identify the final word in one of the two sentences from a visual array of four alternatives given a 4 s time limit (*Figure 2C*, *Figure 2—figure supplement 1*). In 50% of the trials, a visual spatial-attention cue indicated the side of target sentence presentation, the other half of trials were preceded by an uninformative neutral cue.

We extracted individuals' mean accuracy and response speed (calculated as the inverse of reaction time) as key readouts of listening behaviour. On the basis of source-localised 1–8 Hz auditory cortical activity, we further quantified individuals' neural filtering ability as their attention-guided neural tracking of relevant vs. irrelevant speech (*Figure 2D*; see also *Tune et al., 2021* and 'Materials and methods' for details). Our main analyses focus on neural filtering and listening performance averaged across all trials and thereby also across two separate spatial-attention conditions. This choice allowed us to most directly probe the trait-like nature and relationships of neural filtering. It was additionally supported by our previous observation of a general boost in behavioural performance with stronger neural filtering, irrespective of spatial attention (*Tune et al., 2021*).

We follow a three-step analysis strategy to address our specific research questions. First, we provide a largely descriptive overview of the observed average 2-year change per studied domain. Second, we follow up on this fundamental analysis with a causal mediation analysis (*Imai et al., 2010*) and single-trial mixed-effect model analysis geared to assess the longitudinal stability of our recently reported effects of age, hearing acuity, and neural filtering on listening task performance. Third, we integrate and extend the first two analysis perspectives in a joint LCSM (*McArdle, 2009*) to most directly probe the role of neural filtering ability as a predictor of future attentive listening ability.

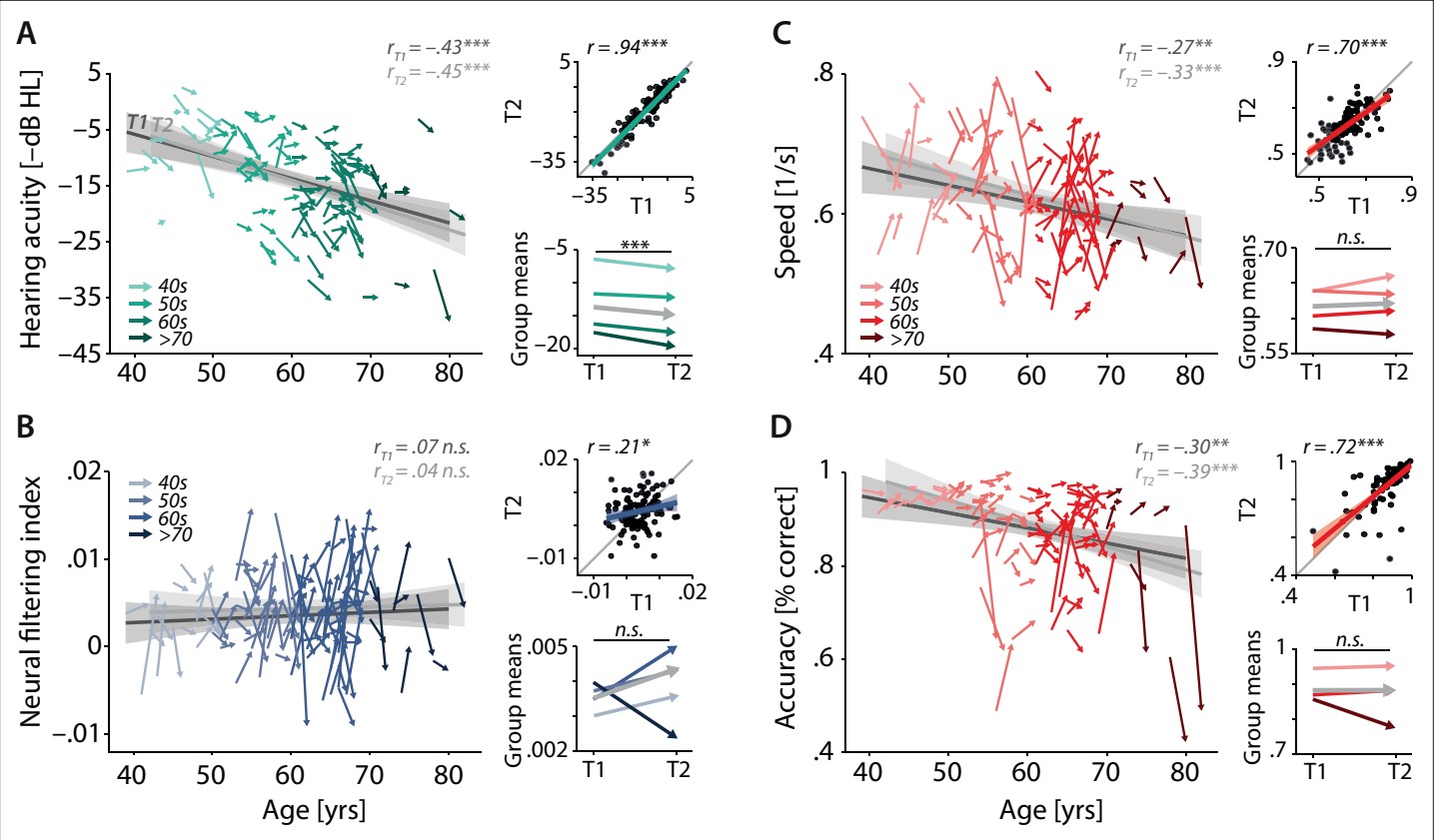

**Figure 3.** Characterising cross-sectional and longitudinal change along the auditory sensory (**A**), neural (**B**), and behavioural (**C, D**) domain. For each domain, coloured vectors (colour coding four age groups for illustrative purposes, only) in the respective left subpanels show an individual's change from T1 to T2 (*N* = 105), along with the cross-sectional trend plus 95% confidence interval (CI) separately for T1 (dark grey) and T2 (light grey). Top right subpanels: correlation of T1 and T2 as a measure of test–retest reliability along with the 45° line (grey) and individual data points (black circles). Bottom right panels: mean longitudinal change per age group (coloured vectors) and grand mean change (grey). Note that accuracy is expressed here as proportion correct for illustrative purposes, but was analysed logit-transformed or by applying generalised linear models.

The online version of this article includes the following figure supplement(s) for figure 3:

**Figure supplement 1.** Observed and projected hearing loss progression.

Addressing our key change-related research questions at the latent rather than the manifest level supersedes the manual calculation of notoriously noisy differences scores and effectively removes the influence of each metric's reliability on the estimation of change-related relationships (*McArdle, 2009*; *Kievit et al., 2018*; *McArdle and Nesselroade, 1994*).

### Listening performance remains stable despite decreased hearing acuity

In a first analysis (*Figure 3*), we characterised how hearing acuity, neural filtering, and listening performance change across the middle to older adult lifespan. Additionally, we analysed longitudinal change from timepoint 1 (T1) to timepoint 2 (T2). We used the same linear mixed-effect models to test cross-sectional effects of age and longitudinal changes with time. We additionally quantified each measure's test–retest reliability as their T1–T2 Pearson's correlation.

Note that throughout the article and all analyses, we reversed the sign of pure-tone average (PTA) values to express them as an index of hearing acuity rather than hearing loss (i.e. higher values indicating better acuity). Similarly, for more intuitive interpretation, accuracy is visually presented as mean proportion correct but was logit-transformed for all statistical analyses to satisfy model assumptions.

As expected, hearing acuity decreased linearly with increasing age (*Figure 3A*, β = –3.4, standard error [SE] = 0.71, p<0.001) and on average by 1.2 dB from T1 to T2 (β = –1.18, SE = 0.27, p<0.001; mean$_{T1}$: –13.72 dB HL [SD: 7.8]; mean$_{T2}$: –14.90 dB HL [8.3]). The effect of age did not change with time (age × timepoint β = –0.35, SE = 0.28, p=0.21). Assuming constant individual

progression rates, this observed change corresponds to a projected average decrease in hearing acuity per decade of –6.3 (SD: 15.3) dB HL (*Figure 3—figure supplement 1*). The magnitude of observed and projected hearing loss progression is well in line with recent large-sample reports (*Linssen et al., 2014*; *Rigters et al., 2018*; *Wiley et al., 2008*). Measurements of hearing acuity showed high test–retest reliability ($r = 0.94$, p<0.001), underscoring the high fidelity of our audiological assessment.

In line with known deleterious effects of age, both behavioural outcomes (response speed and accuracy) declined with increasing age, and did so to a similar degree in T1 and T2 (*Figure 3C and D*; speed: $\beta = -0.02$, SE = 0.01, p=0.004; accuracy: $\beta = -0.23$, SE = 0.07, p=0.0001; timepoint × age p>0.12). At the same time and contrary to our expectations, average performance levels remained stable from T1 to T2 (speed: $\beta = 0.004$, SE = 0.01, p=0.44; $mean_{T1}$: 0.62 $s^{-1}$ [0.08]; $mean_{T2}$: 0.62 $s^{-1}$ [0.08]; accuracy: $\beta = 0.04$, SE = 0.05, p=0.36, $mean_{T1}$: 0.88% [0.09]; $mean_{T2}$: 0.88% [0.11]). Accuracy and response speed showed moderately high test–retest reliability (speed: $r = 0.70$, p<0.001; accuracy: $r = 0.72$, p<0.001).

The analysis of change in neural filtering revealed that its strength varied independently of age at both timepoints (*Figure 3B*, $\beta = 0.0003$, SE = 0.0005, p=0.48; timepoint × age $\beta = -0.0002$, SE = 0.0006, p=0.79), confirming our previously reported T1 results (*Tune et al., 2021*). As shown in *Figure 3* (bottom-left panel), magnitude and direction of observed longitudinal change are highly variable across individuals and age groups, and we did not find evidence of any systematic group-level change from T1 to T2 ($\beta = 0.001$, SE = 0.001, p=0.16). In addition, individual neural filtering strength correlated only weakly across time ($r = 0.21$, p=0.03).

We also assessed the reliability of two established neural traits using resting-state EEG from the same recording sessions: the individual alpha frequency (IAF) (*Corcoran et al., 2018*) and the slope of 1/f neural noise (*Donoghue et al., 2020*; *Voytek et al., 2015*). As expected, both metrics showed high test–retest reliability (IAF: $r = 0.83$, p<0.001; 1/f slope: $r = 0.78$, p<0.001). These findings provide a reference level on reliability, demonstrating that the weak reliability of the neural filtering metric is not due primarily to differences in EEG signal quality across sessions.

The temporal instability of neural filtering challenges its status as a candidate trait-like neural marker of attentive listening ability. At the same time, irrespective of the degree of reliability of neural filtering itself, across individuals it may still be reliably linked to the behavioural outcome (*Figure 1*). This will be addressed next.

## Neural filtering reliably supports listening performance independent of age and hearing status

On the basis of the full (T1 and T2) dataset, we aimed to replicate our key T1 results and test whether the previously observed between-subjects brain–behaviour relationship would hold across time: we expected an individual's neural filtering ability to impact their listening performance (accuracy and response speed) independently of age or hearing status (*Tune et al., 2021*). Given the moderately strong correlation of age and hearing acuity ($r = -0.43$; p<0.001; *Figure 2B*), we employed causal mediation analysis to model the direct as well as the hearing-acuity-mediated effect of age on the behavioural outcome (*Imai et al., 2010*). To formally test the stability of direct and indirect relationships across time, we used a moderated mediation analysis. In this analysis, the inclusion of interactions by timepoint tested whether the influence of age, sensory acuity, and neural filtering on behaviour varied significantly across time.

Our expectations on the direct relationships were indeed borne out by the data: higher age was associated with poorer hearing ability ($\beta = -0.43$, SE = 0.09, p<0.001) and listening performance (speed: $\beta = -0.33$, SE = 0.06, p<0.001; accuracy: $\beta = -0.26$, SE = 0.06, p<0.001). Better hearing ability, on the other hand, boosted accuracy but not response speed (accuracy: $\beta = 0.30$, SE = 0.1, p=0.003; speed: $\beta = -0.06$, SE = 0.1, p=0.56). These direct effects remained stable from T1 to T2 (all interactions by timepoint p>0.56; all log Bayes factors [$logBF_{01}$] >2.5).

Age also impacted accuracy indirectly: the total effect of age was partially mediated via its detrimental effect on hearing acuity (average causal mediation effect [ACME], $\beta = -0.12$, SE = 0.04, p<0.001). We did not find evidence for an analogous indirect effect on speed (ACME: $\beta = 0.008$, SE = 0.03, p=0.77). Again, the hearing-acuity-mediated effect of age on accuracy did not change from T1 to T2 as evidenced by moderated mediation analysis (interaction by timepoint p=0.73).

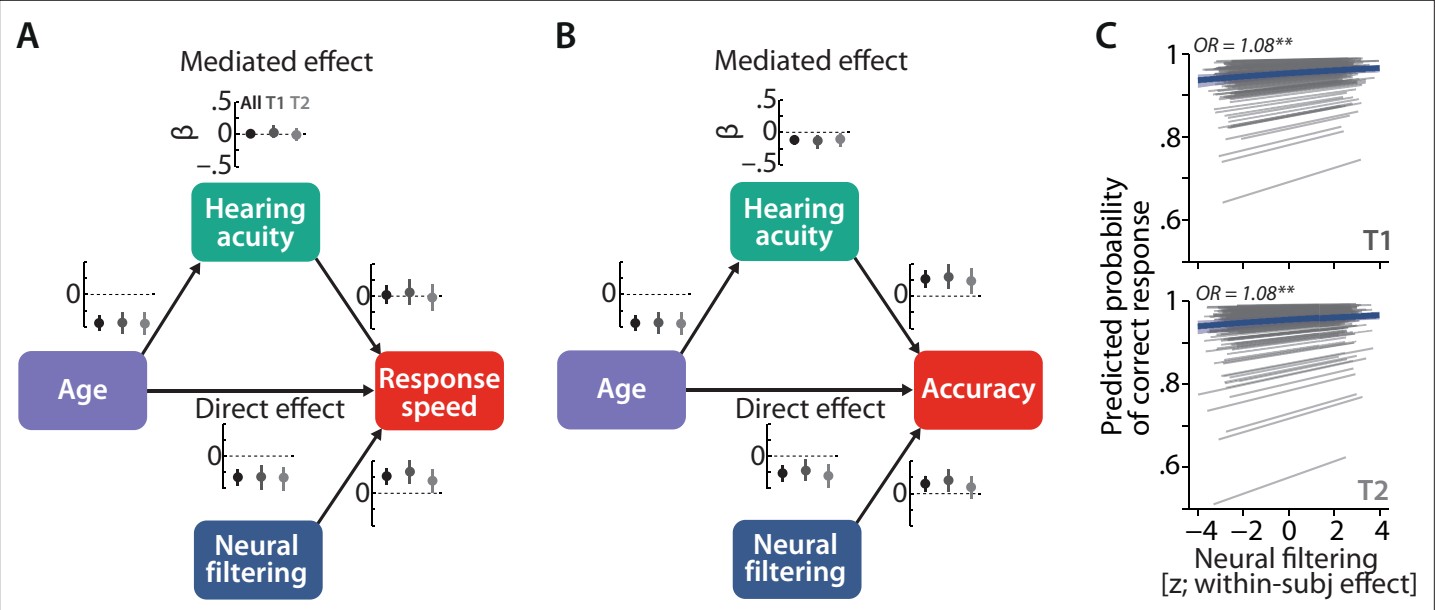

**Figure 4.** Longitudinal stability of sensory and neural determinants of listening behaviour. Causal mediation analysis of age, hearing acuity, and neural filtering on response speed (**A**) and logit-transformed accuracy (**B**) of N = 105 individuals. Graphs next to each path indicate standardised coefficients plus 95% confidence interval (CIs) separately for the full dataset (black), T1 (dark grey), and T2 (light grey). We did not find any significant modulation of the observed effects with time (see text for results). (**C**) Neural filtering strength was found to be predictive of accuracy at the single-trial level at both T1 (top) and T2 (bottom). Grey lines show individual effects, while blue thick line shows the group-level fixed effect along with 95% CI. OR = odds ratio, **p<0.01.

Speaking to the robustness of our previous results, we observed the beneficial effect of stronger neural filtering fidelity on both measures of listening performance (accuracy: $\beta = 0.21$, SE = 0.09, p=0.02; speed: $\beta = 0.33$, SE = 0.09, p<0.001). Note that the magnitude of this direct brain–behaviour effect is comparable to that of the direct effect of age. Alternative models that included indirect, neural filtering-mediated paths from either age or hearing acuity to behaviour did not reveal any significant mediation effects.

Most importantly, the longitudinal stability of the observed direct brain–behaviour link was further supported by the absence of any significant changes with time (interactions by timepoint; all p>0.28, all $\log BF_{01} > 2.1$).

In our previous T1 analysis (N = 155) (*Tune et al., 2021*), we had found evidence for the here analysed brain–behaviour link at two different levels of observation: (i) at the trait level—individuals with overall stronger neural filtering also performed better overall—and (ii) at the state level—stronger neural filtering in a given trial raised the chances of responding correctly. Aiming at replication of the state-level (i.e. within-participant) relationship, we ran a single-trial linear mixed-effect model analysis on our longitudinal N = 105 sample. This analysis utilised single-trial data of both T1 and T2.

Lending credibility to our previous results, stronger single-trial neural filtering was associated with higher listening success at both T1 and T2 (logistic mixed-effect model; within-participant effect of neural filtering on accuracy: odds ratio [OR] = 1.08, SE = 0.02, p<0.001; interaction neural filtering × timepoint: OR = 0.99, SE = 0.03, p=0.82; *Figure 4C*).

## Accuracy is longitudinally stable but speed and neural filtering increase at T2

Having established the longitudinal stability of the beneficial impact of intact neural filtering on listening performance, we turned to our final, most comprehensive analysis.

In an LCSM, in its bivariate form sometimes termed a parallel process model (*McArdle, 2009*; *Kievit et al., 2018*), we connected the neural and behavioural domain. This allowed us to most directly probe the potential role of neural filtering as a precursor of behavioural changes. Specifically, we asked: (i) Is an individual's baseline (T1) level of neural filtering ability predictive of their 2-year

change in behaviour? (ii) Are individual differences in longitudinal dynamics in the behavioural domain associated with those in the neural domain?

As a technical note, it is worth reiterating that in the present data the highly variable, weakly reliable surface measure of neural filtering was nonetheless robustly connected to the behavioural outcome (see above and *Figure 4*). It is in such scenarios that the LCSM framework comes with particular methodological benefits: by expressing individuals' T1 and T2 levels, as well as their T1–T2 change as latent variables instead of manifest indicators, these types of models circumvent the calculation of notoriously unreliable noisy difference scores. They also avoid potential regression to the mean due to random errors (*Kievit et al., 2018*; *McArdle and Nesselroade, 1994*). Instead, the measurement errors of both, latent variables and their associated indicators, are explicitly modelled and thus effectively removed from the estimates of individual differences and relationships of interest (*McArdle, 2009*).

Accordingly, for our metrics, we estimated T1 and T2 latent variables of behavioural and neural filtering from two manifest indicators each. These indicators were the average of each metric across the first and second half of the experiment, respectively (see 'Materials and methods' for details). For all measurement models, the standardised factor loadings were significant (all ps<0.05; all standardised $\lambda > 0.55$). The assumption of strict factorial invariance across time could be maintained for all models (all $\Delta \chi^2_{(df = 1)} < 3.4$, all p>0.07).

We then constructed univariate LCSMs to test for significant mean change in each metric from T1 to T2 while adjusting for their respective baseline (T1) level. The univariate models had acceptable (speed: $\chi^2_{(df = 5)} = 9.7$, p=0.085, comparative fit index [CFI] = 0.988, root mean square error of approximation [RMSEA] = 0.094) to excellent fit (accuracy: $\chi^2_{(df = 5)} = 5.1$, p=0.40, CFI = 0.999, RMSEA = 0.016; neural filtering: $\chi^2_{(df = 5)} = 0.6$, p=0.99, CFI = 1, RMSEA = 0) according to established indices (*Raykov and Marcoulides, 2006*).

On average, listening task accuracy remained stable ($b_0 = 0.053$, SE = 0.045, $\Delta \chi^2_{(df = 1)} = 1.33$, p=0.25). Response speed, on the other hand, showed a significant mean increase over time ($b_0 = 0.13$, SE = 0.03, $\Delta \chi^2_{(df = 1)} = 12.79$, p<0.001; *Figure 5A*). Similarly, the model of neural filtering showed a (marginally) significant mean increase ($b_0 = 0.24$, SE = 0.11, $\Delta \chi^2_{(df = 1)} = 2.98$, p=0.08; *Figure 5A*).

As a proof of principle, we extracted latent factor scores for T1/T2 neural filtering and response speed from their respective univariate LCSM. By correlating T1 and T2 factor scores per domain, we show how the explicit modelling of measurement errors helps to improve the test–retest reliability compared to that observed at the level of manifest variables (see insets in *Figure 5A*; neural filtering: *r* = 0.65, p<0.001, response speed: *r* = 0.75, p<.001).

## Baseline neural functioning does not predict future change in listening behaviour

Based on the univariate results, we then connected the two metrics that showed a significant mean change, namely speed and neural filtering, in a bivariate model of change (*Figure 5B*; see also *Figure 5—figure supplement 1* for full model details).

In line with our hypotheses, we modelled the longitudinal impact of T1 neural functioning on the change in speed and tested for a change–change correlation. Since the analyses conducted up to this point have either directly shown or have suggested that longitudinal change per domain may be affected by age, we included individuals' age at T1 as a time-invariant covariate in the final model. We modelled the influence of age on neural and behavioural functioning at T1 but also on individual change per domain. By accounting for linear effects of age on longitudinal change, we also minimise its potential impact on the estimation of change–change relationship of interest. Note that we refrained from fitting separate models per age group due to both limited and different number of data points per age group. The model fit the data well ($\chi^2_{(df = 27)} = 25.65$, p=0.54, CFI = 1, RMSEA = 0, 95% CI [0.07]).

Having ensured factorial invariance and goodness of fit, we can confidently interpret the estimates of individual differences and bivariate relationships that speak to our specific research questions. Crucial to our change-related inquiries, we observed reliable variance (i.e. individual differences) in the longitudinal change in both speed (standardised estimate $\phi = 0.88$, SE = 0.07, $\Delta \chi^2_{(df = 5)} = 94.07$, p<0.01) and neural filtering ($\phi = 0.81$, SE = 0.14, $\Delta \chi^2_{(df = 4)} = 25.64$, p<.001).

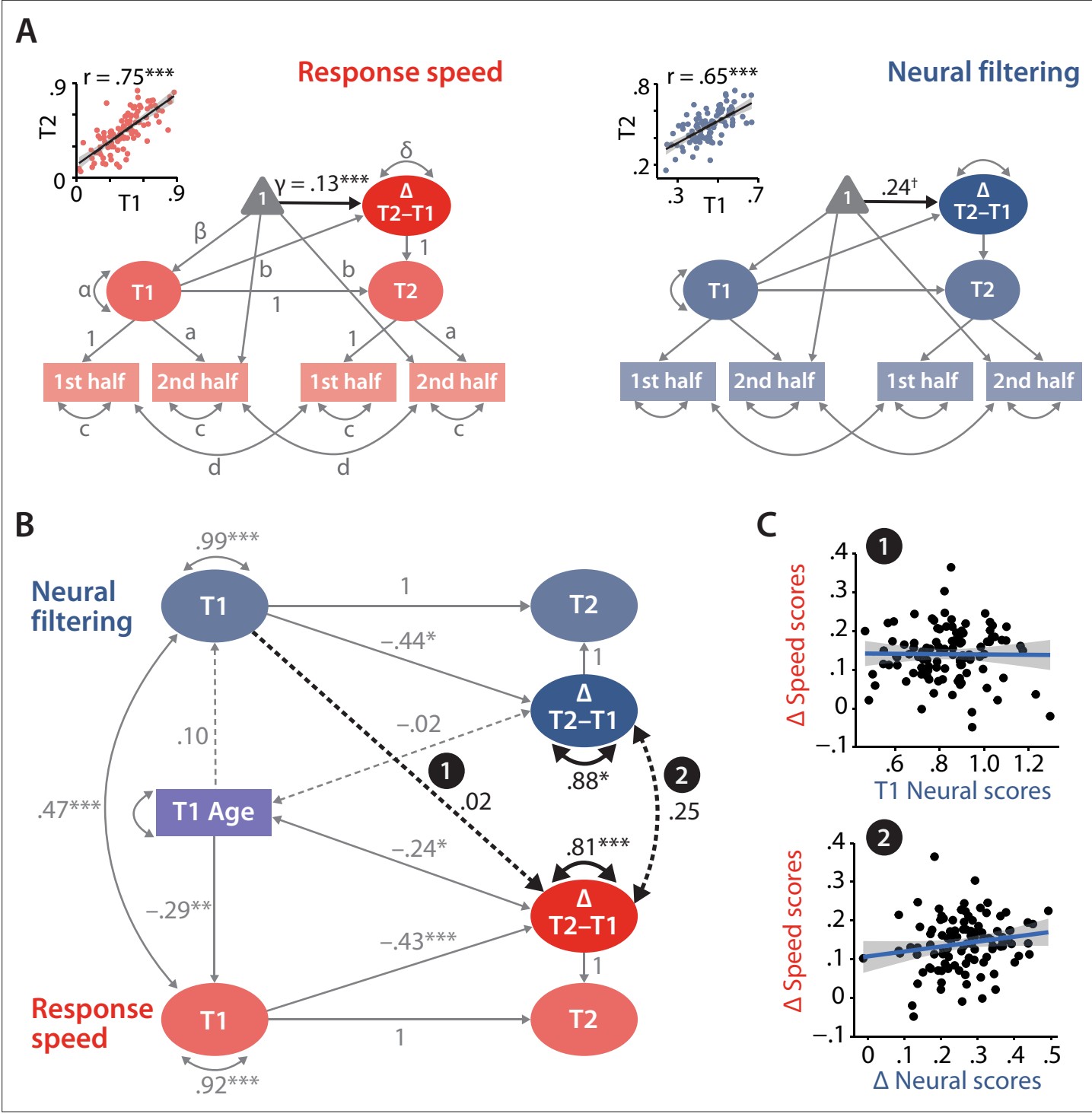

**Figure 5.** Modelling of univariate and bivariate change. (**A**) Univariate latent change score models (LCSM) for response speed (left) and neural filtering (right). All paths denoted with Latin letters refer to freely estimated but constrained to be equal parameters of the respective measurement models. Greek letters refer to freely estimated parameters of the structural model. Highlighted in black is the estimated mean longitudinal change from T1 to T2. Scatterplots in the top-left corner illustrate how capturing T1 and T2 neural and behavioural functioning as latent factors improves their respective test–retest reliability. (**B**) LCSM relating 2-year changes in neural filtering strength to changes in response speed. Black arrows indicate paths or covariances of interest. Solid black arrows reflect freely estimated and statistically significant effects, while dashed black arrows reflect non-significant effects. All estimates are standardised. Grey arrows show paths that were freely estimated or fixed as part of the structural model but that did not relate to the main research questions. For visual clarity, manifest indicators of the measurement model and all symbols relating to the estimated mean structure are omitted but are identical to those shown in panel (**A**). \*\*\*p<0.001, \*\*p<0.01, \*p<0.05, †p=0.08. (**C**) Scatterplots of model-predicted factor scores that

*Figure 5 continued on next page*

Figure 5 continued

refer to the highlighted paths in panel (**B**). Top panel shows that baseline-level neural filtering did not predict 2-year change in behavioural functioning, while bottom panel shows the absence of a significant change–change correlation. All panels include data of $N$ = 105 individuals.

The online version of this article includes the following figure supplement(s) for figure 5:

**Figure supplement 1.** Full bivariate latent change score model of response speed and neural filtering.

**Figure supplement 2.** Change in neural filtering does not predict change in response speed.

Individuals' baseline levels of both speed and neural filtering strength were predictive of their respective longitudinal change: individuals with relatively strong neural filtering or fast responses at T1 showed a smaller increase from T1 to T2, possibly indicating ceiling effects (speed: $\beta$ = –0.43, SE = 0.12, $\Delta \chi^2_{(df = 1)}$ = 8.75, p=0.003; neural filtering: $\beta$ = –0.44, SE = 0.16, $\Delta \chi^2_{(df = 1)}$ = 4.17, p=0.04).

We also observed that participants' age at T1 covaried with the individual degree of change in speed but not with that in neural filtering: The older a participant at T1, the smaller their longitudinal increase in speed (speed: $\phi$ = –0.24, SE = 0.11, $\Delta \chi^2_{(df = 1)}$ = 4.38, p=0.037; neural filtering: $\phi$ = –0.02, SE = 0.14, $\Delta \chi^2_{(df = 1)}$ = 0.02, p=0.89).

Importantly, however, an individual's latent T1 level of neural filtering strength was not predictive of the ensuing latent T1–T2 change in response speed ($\beta$ = 0.02, SE = 0.16, $\Delta \chi^2_{(df = 1)}$ = 0.02, p=0.90). We did not have any a priori hypotheses on the influence of T1 speed on the individual T1–T2 change in neural filtering. Still, in a control analysis that freely estimated the respective path, we found that an individual's latent T1 level of response speed was not predictive of the ensuing latent T1–T2 change in neural filtering ($\beta$ = –0.11, SE = 0.21, $\Delta \chi^2_{(df = 1)}$ = 0.31, p=0.58).

## Neural filtering ability and listening behaviour follow independent developmental trajectories in later adulthood

Finally, we turn to the last piece in our investigation where we address the question of whether individual differences in the neural and behavioural longitudinal change are connected. In other words: Are the contemporaneous changes along the two studied domains correlated or do they occur largely independently of one another?

Change score modelling revealed that longitudinal change in the neural and the behavioural domain occurred largely independent of one another despite their systematic relationship within each separate measurement timepoint ($\phi$ = 0.25, SE = 0.15, $\Delta \chi^2_{(df = 1)}$ = 2.74, p=0.1). In other words, those individuals who showed the largest change in neural filtering were not necessarily the ones who also changed the most in terms of their behavioural functioning (see **Figure 5C**, bottom panel, and **Figure 5—figure supplement 2**).

## Control analyses: The weak correlation of behavioural and neural change is robust against different quantifications of neural filtering

Taken together, our main analyses revealed that inter-individual differences in behavioural change could only be predicted by baseline age and baseline behavioural but not neural functioning, and did not correlate with contemporaneous neural changes.

However, one could ask in how far core methodological decisions taken in the current study, namely our focus on (i) the *differential* neural tracking of relevant vs. irrelevant speech as proxy of neural filtering, and (ii) on its trait-level characterisation that averaged across different spatial-attention conditions may have impacted these results. Specifically, if the neural filtering index (compared to the neural tracking of attended speech alone) is found to be less stable generally, would this also impact the chances of observing a systematic change–change relationship? Relatedly, did the analysis of neural filtering across all trials underestimate the effects of interest?

To evaluate the impact of these consideration on our main findings, we conducted two additional control analyses. First, we repeated the main analyses using the neural filtering index (and response speed) averaged across selective-attention trials, only. Second, we repeated the main analyses focused on the neural tracking of attended speech only, again averaged across selective-attention trials.

As shown in **Figure 6**, taken together, the control analyses provide compelling empirical support for the robustness of our main results: linking response speed and neural filtering under selective attention strengthened their relationship at T1 ($\phi$ = 0.54, SE = 0.15, $\Delta \chi^2_{(df = 1)}$ = 2.74, p=0.1; **Figure 6B**;

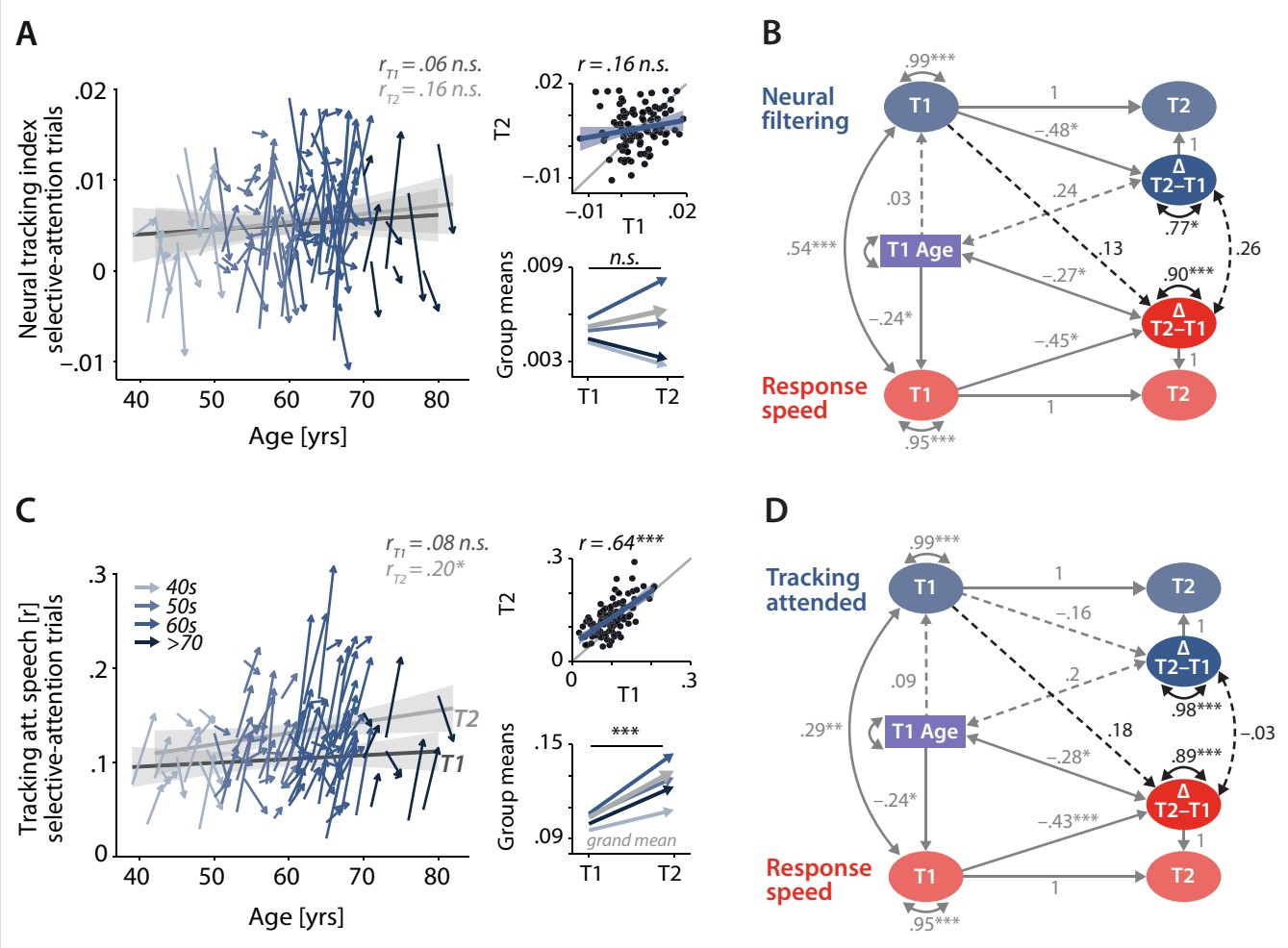

**Figure 6.** Control analyses corroborate the independence of neural and behavioural trajectories under selective attention. Cross-sectional and longitudinal change in neural filtering (**A**) and neural tracking of attended speech (**C**) averaged across selective-attention trials, only. Coloured vectors (colour coding four age groups for illustrative purposes, only) in the left subpanels show individual T1–T2 change along with the cross-sectional trend plus 95% confidence interval (CI) separately for T1 (dark grey) and T2 (light grey). Top right: correlation of T1 and T2 as a measure of test–retest reliability along with the 45° line (grey) and individual data points (black circles). Bottom right: mean longitudinal change per age group and grand mean change (grey). (**B, D**) Latent change score model (LCSM) relating 2-year changes in neural filtering (**B**) /neural tracking (**D**) strength to changes in response speed. Black arrows show the paths or covariances of interest that were freely estimates, while grey arrows show paths that were freely estimated or fixed as part of the structural model but did not relate to the main research questions. Solid arrows indicate statistically significant effects, while dashed arrows reflect non-significant paths. All estimates are standardised. ***$p<0.001$, **$p<0.01$, *$p<0.05$. All panels include data from $N = 105$ individuals.

The online version of this article includes the following figure supplement(s) for figure 6:

**Figure supplement 1.** Cross-sectional and longitudinal change in accuracy and response speed averaged across selective-attention trials, only.

**Figure supplement 2.** Cross-sectional and longitudinal change in neural tracking of ignored speech averaged across selective-attention trials, only.

see also *Figure 6—figure supplement 1*) but did not yield any significant effects for the influence of T1 neural filtering on behavioural change (β = 0.13, SE = 0.21, $\Delta \chi^2_{(df = 1)}$ = 0.43, p=0.51), or for the relationship of neural and behavioural change (φ = 0.26, SE = 0.14, $\Delta \chi^2_{(df = 1)}$ = 3.1, p=0.08; please note the close correspondence to path estimates reported in *Figure 5*).

The second control analysis revealed a substantially higher manifest-level test–retest reliability of neural tracking of attended speech (r = 0.65, p<0.001; *Figure 6C*, see also *Figure 6—figure supplement 2* for neural tracking of ignored speech) compared to that of the neural tracking index. However, when linked to longitudinal changes in response speed, this analysis provided even less evidence for systematic change-related relationships: baseline levels of attended-speech tracking did not predict future change in response speed (β = 0.18, SE = 0.11, $\Delta \chi^2_{(df = 1)}$ = 2.73, p=0.10), and changes in neural

and behavioural functioning occurred independently of one another ($\phi$ = –0.03, SE = 0.12, $\Delta \chi^2_{(df = 1)}$ = 0.06, p=0.81).

In sum, the two control analyses provide additional empirical support for the results revealed by our main analysis.

## Discussion

Successfully comprehending speech in noisy environments is a challenging task, particularly for ageing listeners whose hearing ability gradually declines (*Peelle and Wingfield, 2016*). A much-researched neural support mechanism here is attention-guided neural 'tracking' of behaviourally relevant speech signals, as one neural strategy to maintain listening success (*Wöstmann et al., 2020*; *Ding and Simon, 2012*; *Zion Golumbic et al., 2013*; *Obleser and Kayser, 2019*; *Mesgarani and Chang, 2012*; *Power et al., 2012*; *O'Sullivan et al., 2019*; *Gross et al., 2013*).

However, to date, it is unknown whether the fidelity with which an individual implements this filtering strategy does represent a stable neural trait-like marker of individual attentive listening ability. Of direct relevance to any future translational efforts building on neural speech tracking, it is also unknown whether differences in neural filtering strength observed between ageing listeners are predictive at all of how their attentive listening ability will develop in the future. We here have addressed these questions leveraging a new representative prospective cohort sample of healthy middle-aged to older listeners.

Over 2 y from T1 to T2, individuals' hearing ability worsened as expected (*Linssen et al., 2014*; *Rigters et al., 2018*; *Wiley et al., 2008*). Their listening performance, however, stayed stable. In addition, an individual's baseline (T1) neural filtering strength proved to be a strikingly poor indicator of their future (T2) level of neural filtering. On the other hand, bolstering previous results, neural filtering reliably supported listening behaviour within the same session—both at T1 and T2—at two levels of granularity: individuals with generally stronger neural tracking of target vs. distractor speech performed the listening task on average more accurately and faster. They were also more likely to respond correctly in a given trial with relatively stronger neural filtering in that trial.

Crucially, however, momentary states of neural functioning were not predictive of future behavioural change, and the dynamics of longitudinal change at the neural and behavioural level appear to follow largely independent trajectories. Notably, these key findings hold for different definitions of neural-attentional filtering and under different spatial-attention conditions (see 'Control analyses').

### Neural filtering fidelity as a trait-like neural marker of individual attentive listening ability?

In recent years, the enhanced representation of behaviourally relevant sounds via their prioritised neural tracking has been reported in numerous listening studies investigating different acoustic environments, participant populations, and stages of auditory processing (*Tune et al., 2021*; *O'Sullivan et al., 2015*; *Fiedler et al., 2019*; *Kaufman and Zion Golumbic, 2023*; *Kraus et al., 2021*; *Brodbeck et al., 2020*; *Forte et al., 2017*; *Fuglsang et al., 2020*).

This neural signature is commonly interpreted as a neural instantiation of selective auditory attention in the service of successful speech comprehension. Note however that its link to behaviour is not always explicitly established (but see *Tune et al., 2021*; *O'Sullivan et al., 2015*; *Mesgarani and Chang, 2012*; *Orf et al., 2023*; *Schmitt et al., 2022*). Given how robustly the current data show the enhanced neural tracking of attended vs. ignored speech at the group level, the weak reliability of individual neural filtering strength may come as a surprise. At the same time, stronger neural filtering was reliably linked to better behavioural performance within both T1 and T2.

How can these two findings be reconciled? Based on the current and previous results, what may be concluded about the role of neural filtering as a potential neural marker of individual attentive listening ability?

Previous studies on attention-guided neural speech tracking have not provided any direct evidence on the temporal stability of neural filtering nor on its relationship with behaviour. Studies on related neural signatures such as speech-aligned auditory brainstem responses or the entrainment of auditory cortical activity to rhythmic (non-speech) stimulation reported moderate to high reliability (*Song et al., 2011*; *Easwar et al., 2020*; *Cabral-Calderin and Henry, 2022*; *Panela et al., 2024*). However,

these studies have (i) investigated the temporal stability across sessions spaced only days or weeks apart, (ii) focused on younger normal-hearing populations, or (iii) quantified the neural encoding of speech or non-speech stimulation that involved less or no attentional control. In contrast, we here explicitly focused on a definition of neural filtering that incorporated the neural tracking of attended and ignored speech to highlight the dependence of communication success on the attention-guided differential neural encoding of relevant vs. irrelevant input. These differences render a direct comparison to our approach difficult, but there is reason to consider a model-derived, latent representation of neural filtering as employed here the more generalisable metric (*Hertzog and Schaie, 1988*). Leaving the latent-variable framework aside, it is worth emphasising that the neural tracking of attended speech alone proved to be substantially more stable over time, with a retest reliability in the $r_{tt}$ = 0.6 range.

Our current results broadly align with a view of attention-guided neural speech tracking as a form of 'neural entrainment in the broad sense' that reflects a listener's neural attentional state (*Obleser and Kayser, 2019*; *Lakatos et al., 2016*; *Lakatos et al., 2019*).

Under this interpretation, the stable link of neural filtering to listening behaviour at different levels of granularity is noteworthy: an individual's ability to exert top-down selective attention to prioritise the neural encoding of behaviourally relevant information is far from stable but it fluctuates at different time scales (*Schroeder and Lakatos, 2009*; *Schroeder et al., 2010*; *Helfrich et al., 2018*; *Buschman and Kastner, 2015*). This entails that at a longer time scale, here captured by two distinct measurement timepoints, ageing individuals will differ with respect to their overall level of neural filtering and associated listening behaviour. Their level of neural and behavioural functioning will differ from other individuals' levels at the same timepoint but also from their own level at a different timepoint. Moreover, a listener's behavioural outcome at either timepoint is not only shaped by their broad neural attentional state with which they enter a communication situation. It is also critically influenced by short-term fluctuations in neural filtering strength around their current overall level of neural functioning (*Figure 4C*; *Tune et al., 2021*). As our control analyses have revealed, these fluctuations are more pronounced if neural filtering is defined by the tracking of both attended and ignored speech rather than by attended-speech tracking alone.

What does this mean for the potential translational value of neural tracking? Their highly dynamic nature gives neural tracking-based metrics value as online neural indicators of a listener's momentary attentional focus. As such, they could serve as critical neural read-outs in novel brain–computer interfaces such as neurally steered hearing aids (*Van Eyndhoven et al., 2017*; *Ceolini et al., 2020*; *O'Sullivan et al., 2017*). At the same time, however, their non-stationarity over time limits their potential as translational targets for diagnosis and therapeutic intervention (*Panela et al., 2024*; *Gillis et al., 2021*; *Di Liberto et al., 2022*).

## Individual trajectories of listening behaviour cannot be explained by changes within a single domain

As a second central query of the current study, we went beyond the establishment of robust brain–behaviour relationships and directly probed the potency of neural filtering to predict behavioural change over time (*Woo et al., 2017*). We asked whether individual trajectories of listening behaviour could be predicted by past levels of neural filtering or by co-occurring changes in neural filtering.

Past studies have observed enhanced cortical speech tracking in ageing compared to young adults. This suggests a compensatory role of increased speech–brain coupling to counteract the deleterious effect of age or of hearing loss on speech comprehension (*Schmitt et al., 2022*; *Gillis et al., 2022a*; *Presacco et al., 2019*; *Presacco et al., 2016*; *Decruy et al., 2019*; *Decruy et al., 2020*). As a corollary of this relationship, typically observed cross-sectionally, one might expect an individual's neural filtering strength to be connected not only to present but also to future trajectories of listening behaviour.

Such relationships, if shown to be causal, would provide the strongest evidence for the role of neural speech tracking as a neural compensatory mechanism supporting communication success (*Peelle and Wingfield, 2016*). However, when analysed in our longitudinal sample of ageing listeners, irrespective of the precise definition of neural filtering, we did not find evidence for a predictive role of neural filtering despite supporting brain–behaviour links observed within each timepoint. What are the potential reasons for this absent connection (*Figure 5*)?

One obvious explanation, both in statistical and substantive terms, may lie in the low retest reliability of our neural filtering metric as discussed above. Analytically, however, we were able to mitigate this problem by adopting a modelling approach which effectively removes the influence of measurement error (*McArdle and Nesselroade, 1994*; shown in *Figure 5A*). Still, individual change in response speed could only be predicted by an individual's baseline speed and age but not by baseline neural filtering nor by its longitudinal change. Moreover, our control analysis of attended-speech tracking provided additional empirical support for the absence of such a predictive link despite the neural metric's moderate manifest-level reliability. These findings call for a more substantive explanation that transcends methodological details.

While most desirable from a translational perspective and a core quest in the cognitive neuroscience of ageing, predicting change in cognitive functioning, here listening behaviour, from baseline or longitudinal change in brain function or structure is a non-trivial endeavour. Connecting individual trajectories of neural or cognitive functioning goes beyond the establishment of domain-specific age trends (*Lindenberger et al., 2011*; *Tucker-Drob et al., 2022*). It also goes beyond the mere extrapolation of (age-independent) brain–behaviour relationships observed at a given moment (*Boker and Martin, 2018*; *Raz and Lindenberger, 2011*). Indeed, empirical evidence—and to some degree also theoretical grounds—for robust brain–behaviour baseline–change or change–change associations is limited (*Oschwald et al., 2019*).

Most empirical studies reporting such significant cross-domain change–change correlations have in fact connected behavioural change to alterations in brain structure rather than brain function (*Raz et al., 2005*; *McArdle et al., 2004*; *Bender et al., 2016*; *Ritchie et al., 2015*; *Lövdén et al., 2014*; *Persson et al., 2016*). Focusing on structural change may be advantageous: not only can structural feature be quantified more directly and reliably, they also follow systematic age-dependent trajectories, thereby providing clearer causal pathways for ensuing behavioural change (*Bennett and Madden, 2014*; *Grady, 2012*). Still, less than half of the studies testing such cross-domain associations have indeed observed them (*Oschwald et al., 2019*). From the perspective of theoretical models of neurobiological and cognitive ageing (*Reuter-Lorenz and Park, 2014*; *Cabeza et al., 2018*; *Stern et al., 2019*), the absence of correlated trajectories of neural and cognitive functioning may indeed be the more expected result. These models highlight the multifaceted nature of healthy cognitive ageing in which environmental variables, neurobiology, and cognition are dynamically interrelated (*Freund et al., 2013*). Neural compensatory mechanisms, such as the neural filtering correlate targeted here, are thought to offset structural decline but are themselves influenced by a number of factors. This leads to increased inter-individual variability that may circumvent the emergence of group-level correlated change relationships.

Importantly, the behavioural outcome of interest, that is, the speed and accuracy with which an ageing individual solves a difficult listening situation, involves the orchestration of different perceptual and cognitive processes (*Anderson et al., 2013*; *Peelle, 2016*). We here focused on one candidate neurobiological implementation of an auditory attentional filter to help explain inter-individual differences in listening behaviour and its lifespan trajectories. Yet, ageing individuals may rely on different alternative neural or cognitive strategies (*Tune et al., 2018*). A complete understanding of inter-individual differences in listening behaviour in ageing adults will therefore depend on a number of different factors among which the attention-modulated tracking of relevant speech constitutes one, potentially necessary, but not sufficient neural correlate (*Wöstmann et al., 2020*; *Obleser and Kayser, 2019*; *Tune et al., 2021*; *Gillis et al., 2022b*; *Strauß et al., 2014*).

Not least, there are a number of methodological choices that might constrain the conclusions afforded by our current study. First, the current study was limited to two distinct timepoints spaced only 2 y apart. This limits the ability to model linear as well as non-linear dynamics of change. Second, it also does not allow the separation of distinct patterns of change co-occurring at the same time: one continuous, constant change with age along with a separate process in which relative change is proportional to the level observed at prior timepoints (*Jacobucci et al., 2019*). Third, we here focus on a single dichotic listening task from which both neural and behavioural functioning are derived. We therefore cannot assess whether the observed pattern would generalise to other listening tasks. Lastly, denser sampling across a longer time interval would have also increased statistical power to detect correlated change (*Rast and Hofer, 2014*). It would have also allowed to more directly test

hypotheses on causal pathways by which change in the neural domain should precede change in the behavioural domain.

The conclusion stands, though, that individual trajectories in listening behaviour cannot be explained by longitudinal change along a single dimension. Instead, a better understanding of the influences shaping individual listening behaviour across the adult lifespan will critically rely on uncovering the relative contribution and age-dependent dynamics of sensory, neural, and cognitive factors.

## Conclusion

The results presented here support the role of attention-guided neural filtering as a readout of an individual's neural attentional state. At the same time, the state-like nature of neural tracking-based metrics limits their translational potential as predictors of longitudinal change in listening behaviour over middle to older adulthood. Our data caution against explaining audiology-typical listening performance solely from changes in aspects of neural functioning as listening behaviour and neural filtering ability follow largely independent developmental trajectories. Our results critically inform translational efforts aimed at the preservation and restoring of speech comprehension abilities in ageing individuals.

## Materials and methods
### Data collection

The analysed data are part of a large-scale longitudinal study on the neural and cognitive mechanisms supporting adaptive listening behaviour in a prospective cohort of healthy middle-aged and older adults ('The listening challenge: How ageing brains adapt (AUDADAPT)'; https://cordis.europa.eu/project/rcn/197855_en.html). This project encompassed the collection of different demographic, audiological, behavioural, and neurophysiological measures across initially two timepoints spaced approximately 2 y apart. The analyses carried out on the data aim at relating adaptive listening behaviour to changes in different neural dynamics. Given the longitudinal nature of the current study, all procedures concerning data collection, as well as EEG recording and analysis, are identical to those detailed in our recently published analysis of T1 data using the same experimental paradigm (*Figure 2—figure supplement 1*; *Tune et al., 2021*).

### Participants and procedure

We here report on a total *N* = 105 right-handed German native speakers (median age at T2 63 y; age range 39–82 y; 61 females) who underwent audiological, behavioural, and EEG assessment at two separate timepoints. On average, the measurement timepoints were spaced 23.2 (± SD 4.0) mo apart.

At T1, we had screened a total of *N* = 184 participants. Included participants had normal or corrected-to-normal vision, and did not report any neurological, psychiatric, or other disorders. They were also screened for mild cognitive impairment using the German version of the 6-Item Cognitive Impairment Test (6CIT [*Jefferies and Gale, 2012*] and the MoCA [*Nasreddine et al., 2005*]). Only participants with normal hearing or age-adequate mild-to-moderate hearing loss were included (*Figure 2B* for individual audiograms at T1). Handedness was assessed using a translated version of the Edinburgh Handedness Inventory (*Oldfield, 1971*). As a result of the initial screening procedure, 17 participants were excluded prior to EEG recording due to a medical history or non-age-related hearing loss. Three participants dropped out of the study prior to EEG recording and an additional nine participants were excluded from analyses after EEG recording due to incidental findings after structural MR acquisition (*N* = 3) or due to EEG data quality issues (*N* = 9). Again, all detailed criteria can be found in *Tune et al., 2021*.

At T2, *N* = 115 participants returned for follow-up measurements. All individuals passed the repeat screening procedures identical to those at T1. Ten participants had to be excluded from the analyses reported here: three participants had dropped out prior to EEG recording, three participants were excluded due to EEG data quality issues, and four participants because their EEG data had been excluded at T1. This resulted in a final longitudinal sample of *N* = 105 individuals.

Dropout at T2 could not be predicted from participants' T1 age, hearing loss, behavioural performance (accuracy, speed), or neural filtering strength (all p>0.13). This indicates that compared to the

full T1 cohort reported on in previous studies (*Tune et al., 2021*; *Alavash et al., 2021*) our reduced longitudinal sample was not biased in terms of sensory, cognitive, or neural functioning.

At each measurement timepoint, participants underwent detailed pure-tone and speech audiometric measurements, along with an extensive battery of cognitive tests and personality profiling (see *Tune et al., 2018* for details). On a separate day, we recorded participants' EEG during rest (5 min each of eyes-open and eyes-closed measurements) followed by six blocks of the same dichotic listening task (see *Figure 2C* and *Tune et al., 2021* for details).

Participants gave written informed consent and received financial compensation (10€ per hour). Procedures were approved by the ethics committee of the University of Lübeck and were in accordance with the Declaration of Helsinki.

## Dichotic listening task

At each timepoint, participants performed a previously established dichotic listening task (*Alavash et al., 2019*). We provide full details on trial structure, stimulus construction, recording, and presentation in our previously published study on the first (*N* = 155) wave of data collection (but see also *Figure 2—figure supplement 1*; *Tune et al., 2021*).

In short, in each of the 240 trials, participants listened to two competing, dichotically presented five-word sentences spoken by the same female speaker. They were probed on the sentence-final noun in one of the two sentences. Participants were instructed to respond within a given 4 s time window beginning with the onset of a probe screen showing four alternatives. They were not explicitly instructed to respond as quickly as possible. The probe screen showed four alternative words presented either on the left or right side of the screen, indicating the probed ear. Two visual cues preceded auditory presentation. First, a spatial-attention cue either indicated the to-be-probed ear, thus invoking selective attention, or did not provide any information about the to-be-probed ear, thus invoking divided attention. Second, but irrelevant to the current study, a semantic cue specified a general or a specific semantic category for the final word of both sentences, thus allowing to utilise a semantic prediction. Cue levels were fully crossed in a 2 × 2 design and presentation of cue combinations varied on a trial-by-trial level. All participants listened to the same 240 sentence pairs at each of the two measurement timepoints. The order of sentence pair presentation was randomised for each participant and at each timepoint.

To account for differences in hearing acuity within our group of participants, all stimuli were presented 50 dB above the individual sensation level.

## EEG recording and analysis

The approach for EEG recording, pre-processing, and subsequent analysis is identical to the procedures carried out for T1 data collection and analysis (*Tune et al., 2021*).

In short, 64-channel EEG data were recorded, cleaned for artefacts using a custom ICA-based pipeline, downsampled to 125 Hz, filtered between 1 and 8 Hz, and cut into single-trial epochs covering the presentation of auditory stimuli. Following source localisation via beamforming, we focused on auditory cortical activity to train and test decoding models of attended and ignored speech using cross-validated regularised regression. Models were trained on selective-attention trials, only, but then also tested on divided-attention trials. As results, we obtained single-trial reconstruction accuracy (Pearson's *r*) estimates as metrics of the degree of attended and ignored neural speech tracking, respectively.

We then calculated a neural filtering index across the entire sentence presentation period. The index quantifies the difference in neural tracking of the to-be-attended and of the to-be-ignored sentence [neural filtering index = $(r_{attended} - r_{ignored})/(r_{attended} + r_{ignored})$], and thus indexes the strength of neural filtering at the single-trial level. Positive values indicate successful neural filtering in line with the behavioural goal.

The EEG analyses were carried out in MATLAB 2016b using the Fieldtrip toolbox (v. 2017-04-28), the Human Connectome Project Workbench software (v1.5), FreeSurfer (v.6.0), and the multivariate temporal response function (mTRF) toolbox (v1.5) (*Crosse et al., 2016*).

## Behavioural and audiological data analysis

We evaluated participants' behavioural performance in the listening task with respect to accuracy and response speed. For the binary measure of accuracy, we excluded trials in which participants failed to

answer within the given 4 s response window ('timeouts'). Spatial stream confusions, that is, trials in which the sentence-final word of the to-be-ignored speech stream were selected, and random errors were jointly classified as incorrect answers. The analysis of response speed, defined as the inverse of reaction time, was based on correct trials only.

We defined participants' hearing acuity as their PTA composed of (air-conduction) hearing thresholds at the frequencies of 0.5, 1, 2, and 4 kHz. Individual PTA values were then averaged across the left and right ear.

## Statistical analysis

For statistical analyses focused on between-participant ('trait-level') effects, behavioural performance metrics and neural filtering index values were averaged across all trials and experimental conditions to arrive at one trait-level estimate per participants. This approach was also motivated by previous results based on T1 data: here, we had observed that stronger neural speech tracking led to overall faster and to more trials with accurate responses irrespective of the specific cue-cue condition (*Tune et al., 2021*). Accuracy was logit-transformed for statistical analysis but expressed as proportion correct for illustrative purposes. Similarly, for more intuitive interpretation, we reversed the sign of PTA values for higher values to correspond to better hearing ability.

All analyses were performed in R (v.4.2.2; *R Development Core Team, 2019*) using the packages lme4 (v.1.1-31; *Bates et al., 2015*), mediation (v4.5.0; *Tingley et al., 2014*), lavaan (v0.6-12; *Rosseel, 2012*), and OpenMx (v2.21.8; *Neale et al., 2016*).

## Linear mixed-effect modelling

As the first step of our three-part analysis approach, we applied general linear mixed-effect models to test for cross-sectional and longitudinal changes in trait-level sensory acuity, neural filtering, and listening performance. These models included age, timepoint, and their interaction as fixed effect regressors and allowed random participant-specific intercepts.

In the second step of the analysis, we also aimed at replicating the previously observed single-trial state-level relationship of neural filtering and accuracy. To this end, we applied a single generalised linear mixed-effect model (binomial distribution, logit link function) on single-trial data of both T1 and T2. This model represents an adapted version of the brain–behaviour model reported in *Tune et al., 2021*. In short, we included all experimental manipulation predictors, as well as age, hearing acuity, and neural filtering metrics. We omitted previously shown to be non-significant higher-order interactions and additionally included interactions by timepoint to directly test for any longitudinal change in the effect of neural filtering on behaviour.

To tease apart state-level (i.e. within-participant) and trait-level (i.e. between-participants) effects, we included two separate neural regressors: for the between-participant effect regressor, we averaged single-trial neural filtering values per individual across all trials. By contrast, the within-participant effect of interest was modelled by the trial-by-trial deviation from the subject-level mean (*Bell et al., 2019*). We included participant- and item-specific random intercepts as well as random slopes for the effect of the spatial-attention cue and the probed ear.

We used deviation coding for categorical predictors and z-scored all continuous regressors. p-Values for individual model terms in general linear mixed-effect models are based on the Satterthwaite approximation for degrees of freedom, and on z-values and asymptotic Wald tests for the generalised linear mixed-effect model of accuracy (*Luke, 2017*).

## Causal mediation analysis

We performed causal mediation analysis to model the direct as well as the hearing-acuity-mediated effect of age on accuracy and response speed (*Imai et al., 2010*). Critically, these models also included a direct (i.e. independent of age and hearing loss) path of neural filtering on behaviour. To formally test the stability of direct and indirect relationships across time, we used a moderated mediation analysis (*Preacher et al., 2007*). In this analysis, the inclusion of interactions by timepoint tested whether the influence of age, sensory acuity, and neural functioning on behaviour varied across time. We z-scored all dependent and independent variables, and estimated the magnitude of direct and mediated effects along with percentile-based confidence intervals on the basis of 1000 replications.

## Bayes factor calculation

To facilitate interpretation of non-significant effects, we calculated the Bayes factor (BF) based on the comparison of Bayesian information criterion (BIC) (*Wagenmakers, 2007*). We report log Bayes factors, with a log $BF_{01}$ of 0 representing equal evidence for and against the null hypothesis; log $BF_{01}$ with a positive sign indicating relatively more evidence for the null hypothesis than the alternative hypothesis, and vice versa. Magnitudes >1 are taken as moderate, >2.3 as strong evidence for either of the alternative or null hypotheses, respectively (*Lee and Wagenmakers, 2014*).

## Latent change score modelling

In the third and final step of our analysis approach, we used structural equation modelling (SEM) to investigate the role of neural filtering as a predictor of behavioural change. We used LCSM (*McArdle, 2009*; *Kievit et al., 2018*) to test (i) whether an individual's T1 neural filtering strength can be predictive of 2-year changes in behaviour, and (ii) whether 2-year changes in neural filtering and listening behaviour are systematically related. All models were specified and fitted with the R-based package lavaan (*Rosseel, 2012*) using maximum likelihood estimation.

To bring all manifest variables onto the same scale while preserving mean differences over time, we first stacked them across timepoint and then rescaled them using the proportion of maximum scale ('POMS') method (*Moeller, 2015*; *Little, 2013*). We assessed model fit using established indices including the $\chi^2$ test, the RMSEA, and the CFI (*Raykov and Marcoulides, 2006*; *Kline, 2023*). Likelihood ratio tests helped us decide whether (i) constraining individual parameters to be equal significantly decreased model fit and (ii) individual parameter estimates were statistically significant. We report standardised parameter estimates.

In a first step, we specified separate unstructured measurement models of accuracy, speed, and neural filtering to establish factorial invariance across time using a series of tests (*Kievit et al., 2018*; *Polk et al., 2022*). Latent variables representing each metric at T1 and T2 were constructed from the observed individual means averaged across the first and second half of the experiment, respectively. Covariances between T1 and T2 latent variables were freely estimated, and residual covariances were set to be equal for the first and second half of the experiment. The factor loading and mean of the first manifest variable were set to 1 and 0, respectively, to ensure model identification (*Little, 2013*). Given our choice of POMS transformation of raw values to preserve mean differences over time, the mean of the second manifest variable had to be freely estimated to avoid model misfit. We sequentially tested for metric (i.e. identical factor loadings), strong (i.e. identical means), and strict (i.e. identical residuals) time invariance using likelihood ratio tests.

Next, for those metrics surviving factorial invariance testing, we constructed separate (i.e. univariate) change score models to test for group-level mean change following the tutorial by *Kievit et al., 2018*. We included a regression path from the T1 latent variable to the latent change variable to test how baseline functioning impacted the degree of longitudinal change. A likelihood ratio test of nested models (with mean change being freely estimated vs. constrained to be 0) determined the significance of group-level mean change.

In the last and final step, we constructed the bivariate LCSM that connected changes in response speed and neural filtering, which both significantly increased over time (see *Figure 5—figure supplement 1* for full model details). We modelled the covariance between the baseline neural filtering and speed, as well as the covariance between neural and behavioural change. In addition, we included a regression path from T1 neural filtering to the T1–T2 change in response speed to directly test the predictive potency of baseline neural functioning to explain behavioural change.

Following the logic of a generalised variance test (*Brandmaier et al., 2018*), we tested for significant inter-individual differences in neural and behavioural change. Note that such reliable variance in change is a necessary prerequisite for testing change–change correlations. In short, per domain (neural, behaviour), a likelihood ratio test compared the full model with a restricted model in which all paths pointing to or connected with the latent change variable are fixed to 0.

Finally, we asked whether neural and behavioural change covaried significantly. We ran a likelihood ratio test on the full model compared to a restricted model in which the covariance parameter was fixed to 0. For visualisation of results, we refit the final model using OpenMx (*Neale et al., 2016*) to predict latent factor scores based on maximum likelihood.

## Acknowledgements

Research was funded by the European Research Council (grant no. ERC-CoG-2014-646696 'Audadapt' awarded to JO). All data were acquired with the help of Franziska Scharata. The authors thank Andreas Brandmaier for helpful comments on the LCSM approach. The comments of three anonymous reviewers helped improve the manuscript.

## Additional information

### Competing interests

Jonas Obleser: Reviewing editor, eLife. The other author declares that no competing interests exist.

### Funding

| Funder | Grant reference number | Author |
|---|---|---|
| European Research Council | ERC-CoG-2014-646696 | Jonas Obleser |

The funders had no role in study design, data collection and interpretation, or the decision to submit the work for publication.

### Author contributions

Sarah Tune, Conceptualization, Data curation, Formal analysis, Investigation, Visualization, Methodology, Writing - original draft, Writing – review and editing; Jonas Obleser, Conceptualization, Resources, Supervision, Funding acquisition, Investigation, Methodology, Project administration, Writing – review and editing

### Author ORCIDs

Sarah Tune http://orcid.org/0000-0001-9022-9965
Jonas Obleser https://orcid.org/0000-0002-7619-0459

### Ethics

Participants gave written informed consent and received financial compensation (10€ per hour). Procedures were approved by the ethics committee of the University of Lübeck (reference no. 15-073) and were in accordance with the Declaration of Helsinki.

Reviewer #1 (Public Review): https://doi.org/10.7554/eLife.92079.3.sa1
Reviewer #2 (Public Review): https://doi.org/10.7554/eLife.92079.3.sa2
Author Response https://doi.org/10.7554/eLife.92079.3.sa3

## Additional files

### Supplementary files
• MDAR checklist

### Data availability

The complete dataset associated with this work including raw data, EEG and behavioural data analysis results, as well as corresponding analysis code are publicly available under https://osf.io/28r57/ (data: https://osf.io/mjpy3/; code: https://osf.io/67har/).

The following dataset was generated:

| Author(s) | Year | Dataset title | Dataset URL | Database and Identifier |
|---|---|---|---|---|
| Tune S, Obleser J | 2024 | AUDADAPT T1 T2 Brain Behaviour | https://doi.org/10.17605/OSF.IO/MJPY3 | Open Science Framework, 10.17605/OSF.IO/MJPY3 |

The following previously published dataset was used:

| Author(s) | Year | Dataset title | Dataset URL | Database and Identifier |
|---|---|---|---|---|
| Tune S, Alavash M, Fiedler L, Obleser J | 2021 | AUDADAPT T1 Brain Behaviour | https://doi.org/10.17605/OSF.IO/G8SD5 | Open Science Framework, 10.17605/OSF.IO/G8SD5 |

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
