## [Editor Report · eLife assessment]

This study provides a **valuable** contribution to understanding the neural mechanisms underlying age-related changes in attention and speech understanding. The large dataset (*N* = 105) provides **convincing** evidence for how speech recognition behaviour and neural tracking of speech separately evolve in about 2 y. The work would be of interest to psychologists, neuroscientists, and audiologists.

---

## [Referee Report · Reviewer #1 (Public Review)]

Summary:

This study investigated behavioural performance on a competing speech task and neural attentional filtering over the course of two years in a group of middle-aged to older adults. Neural attentional filtering was quantified using EEG by comparing neural envelope tracking to an attended vs. an unattended sentence. This dataset was used to examine the stability of the link between behavior and neural filtering over time. They found that neural filtering and behavior were correlated during each measurement, but EEG measures at the first timepoint did not predict behavioural performance two years later. Further, while behavioural measures showed relatively high test-retest reliability, the neural filtering reliability was weak with an r value of 0.21. The authors conclude that neural tracking-based metrics have limited ability to predict longitudinal changes in listening behavior.

Strengths:

This study is novel in its tracking of behavioural performance and neural envelope tracking over time, and it includes an impressively large dataset of 105 participants. The manuscript is clearly written.

Weaknesses:

The weaknesses are minor, primarily concerning how the reviewers interpret their data. Specifically, the envelope tracking measure is often quite low, close to the noise floor, and this may affect test-retest reliability. Furthermore, the trajectories may be affected by accelerated age-related declines that are more apparent in neural tracking than in behaviour.

Comments on revised version:

The authors have satisfactorily addressed my previous comments and they present a strong case for the interpretation of their findings.

---

## [Referee Report · Reviewer #2 (Public Review)]

Summary:

This study examined the longitudinal brain-behaviour link between attentional neural filtering and listening behaviour among a sample of aging individuals. The results based on the latent change score modeling showed that neither attentional neural filtering at T1 nor its T1-T2 change predicted individual two-year listening performance change. The findings suggest that neural filtering and listening behaviour may follow independent developmental trajectories. This study focuses on an interesting topic and has the potential to contribute a better understanding of the neurobiological mechanisms of successful communication across the lifespan.

Strengths:

Although research suggests that speech comprehension is neurally supported by an attention-guided filter mechanism, the evidence on their causal association is limited. This study addresses this gap by testing longitudinal stability of neural filtering as a neural mechanism upholding listening performance, potentially shedding lights on translational efforts aiming at the preservation of speech comprehension abilities among aging individuals.

The latent change score modeling approach is appropriately used as a tool to examine key developmental questions and distinguish the complex processes underlying lifespan development in brain and behaviour with longitudinal data.

Weaknesses:

Although the paper does have strengths in principle, the weaknesses of the paper are that the findings are merely based on a single listening task. Since both neural and behavioral indicators are derived from the same task, the results may be applicable only to this specific task, and it is difficult to extrapolate them to cognitive and listening abilities measured by the other tasks. Therefore, more listening tasks are required to comprehensively measure speech comprehension and neural markers.

The age span of the sample is relatively large. Although no longitudinal change from T1 to T2 was found at the group-level, from the cross-sectional and longitudinal change results (see Figure 3), individuals of different age groups showed different development pattern. Particularly, individuals over the age of 70 show a clear downward trend in both neural filtering index and accuracy. Therefore, different results may be found based on different age groups, especially older groups. However, due to sample limitations, this study was unable to examine whether age has a moderating effect on this brain-behaviour link.

In the Dichotic listening task, valid and invalid cues were manipulated. According to the task description, the former could invoke selective attention, whereas the latter could invoke divided attention. It is possible that under the two conditions, the neural filtering index may reflect different underlying cognitive processes, and thus may differ in its predictive effect on behavioral performance. The author could perform a more in-depth data analysis on indicators under different conditions.

---

## [Author Response]

The following is the authors’ response to the original reviews.

**Reviewer #1 (Public Review):**
Summary:This study investigated behavioural performance on a competing speech task and neural attentional filtering over the course of two years in a group of middle-aged to older adults. Neural attentional filtering was quantified using EEG by comparing neural envelope tracking to an attended vs. an unattended sentence. This dataset was used to examine the stability of the link between behavior and neural filtering over time. They found that neural filtering and behavior were correlated during each measurement, but EEG measures at the first time point did not predict behavioural performance two years later. Further, while behavioural measures showed relatively high test-retest reliability, the neural filtering reliability was weak with an r-value of 0.21. The authors conclude that neural tracking-based metrics have limited ability to predict longitudinal changes in listening behavior.Strengths:This study is novel in its tracking of behavioural performance and neural envelope tracking over time, and it includes an impressively large dataset of 105 participants. The manuscript is clearly written.Weaknesses:The weaknesses are minor, primarily concerning how the reviewers interpret their data. Specifically, the envelope tracking measure is often quite low, close to the noise floor, and this may affect testretest reliability. Furthermore, the trajectories may be affected by accelerated age-related declines that are more apparent in neural tracking than in behaviour.

We thank the reviewer for their supportive assessment of our work. We describe in detail how we have addressed the two main concerns raised here—neural filtering’s low test-retest reliability and differences in age-related behavioural vs. neural change—in our response to the more detailed recommendations below.

To briefly summarise here:

(1) In Figure 5, we now illustrate more transparently how the employed structural equation framework helps to overcome the issue of low test-retest reliability of neural filtering as originally reported.

(2) We include two additional control analyses, one of which relates neural tracking of attended speech (featuring a moderately high T1–T2 correlation of r = .64 even outside of latent modelling) to behavioural change. Importantly, this analysis provides critical empirical support for the apparent independence of neural and behavioural trajectories.

(3) We more clearly describe how the latent-variable modelling strategy accounts for differences in age-related change along the neural and behavioural domain. Moreover, the results of the of 18 additional control analysis also suggest that the absence of a change-change relationship is not primarily due to differential effects of age on brain and behaviour.

**Reviewer #1 (Recommendations For The Authors):**
1. Figure 3:Does the 70-year range reach a tipping point?Is that why neural filtering drops dramatically in this age group, whereas the other groups do not change or increase slightly?This can also be seen with behavioral accuracy to a lesser extent. Perhaps test-retest reliability is affected by accelerated age-related declines in older listeners, as was found for envelope tracking measures in Decruy et al. 2019.

We agree with the reviewer that at first glance the data seem to suggest a critical tipping point in the age range above 70 years. It is important to emphasize, however, that the four age bins were not based on equal number of data points. In fact, the >70 age group included the fewest participants, leading to a less reliable estimate of change. Together with the known observation of increasing interindividual differences with increasing age, the results do not allow for any strong conclusions regarding a potential tipping point. For the same reasons, we used the four age bins for illustrative purposes, only, and did not include them in any statistical modelling.

We did however include chronological age as a continuous predictor in latent change score modelling. Here, we modelled its influence on participants’ T1 neural and behavioural status, as well as its effect on their respective change, thereby accounting for any differential (linear) effects of age on neural vs. behavioural functioning and its change.

On p.14 of the revised manuscript, we now state more clearly that the latent change score model did in fact account for the potential influence of age on the change-related relationships:

"In line with our hypotheses, we modelled the longitudinal impact of T1 neural functioning on the change in speed, and tested for a change-change correlation. Since the analyses conducted up to this point have either directly shown or have suggested that longitudinal change per domain may be affected by age, we included individuals’ age as a time-invariant covariate in the final model. We modelled the influence of age on neural and behavioural functioning at T1 but also on individual change per domain. By accounting for linear effects of age on longitudinal change, we also minimize its potential impact on the estimation of change-change relationship of interest. Note that we refrained from fitting separate models per age group due to both limited and different number of data points per age group."

1. Would good test-retest reliability be expected when the actual values of envelope tracking for attended vs. unattended speech are so low? The investigators address this by including measurement errors in the models, but I am not certain this kind adequately deals with envelope tracking values that are close to the noise floor.

We thank the reviewer for this comment. We addressed the concerns regarding the low re-test reliability of our neural-attentional metric (and its potential impact on observing a systematic changechange relationship) in two separate ways.

The major outcome of these tests is that low re-test reliability of neural tracking is (i) not generally true, and (ii) is not the cause of the main finding, i.e., a low or absent correlations of behavioural vs. neural changes over time.

In more detail, to show how latent change score modelling improves test-retest reliability by explicitly modelling measurement error, we first extracted and correlated T1 and T2 latent factors scores from the respective univariate models of neural filtering and response speed.

Indeed, at the latent level, the correlation of T1–T2 neural filtering was moderately high at r = .65 (compared to r = .21 at the manifest level). The correlation of T1–T2 response speed was estimated as r = .75 (compared to r = .71).

Figure 5A, reproduced below for the reviewer’s convenience, now includes insets quantifying these latent-level correlations over time.

**Author response image 1. sa3fig1:** Modelling of univariate and bivariate change. (**A**) Univariate latent change score models for response speed (left) and neural filtering (right). All paths denoted with Latin letters refer to freely estimated but constrained to be equal parameters of the respective measurement models. Greek letters refer to freely estimated parameters of the structural model. Highlighted in black is the estimated mean longitudinal change from T1 to T2. Scatterplots in the top left corner illustrate how capturing T1 and T2 neural and behavioural functioning as latent factors improves their respective test-retest reliability. (**B**) Latent change score model (LCSM) relating two-year changes in neural filtering strength to changes in response speed. Black arrows indicate paths or covariances of interest. Solid black arrows reflect freely estimated and statistically significant effects, dashed black arrows reflect non-significant effects. All estimates are standardised. Grey arrows show paths that were freely estimated or fixed as part of the structural model but that did not relate to the main research questions. For visual clarity, manifest indicators of the measurement model and all symbols relating to the estimated mean structure are omitted but are identical to those shown in panel A. ***p<.001, **p<.01, *p<.05, p=.08. (**C**) Scatterplots of model-predicted factor scores that refer to the highlighted paths in panel B. Top panel shows that baseline-level neural filtering did not predict two-year change in behavioural functioning, bottom panel shows the absence of a significant change-change correlation.

Second, we ran a control analysis that includes the neural tracking of attended speech in selectiveattention trials rather than the neural filtering index averaged across all trials. The results are shown as part of a new main figure (and two new supplemental figures) reproduced below (see in particular Figure 6, panels C and D).

This analysis serves two purposes: On the one hand, it allows for a more direct evaluation of the actual strength of neural speech tracking as quantified by the Pearson’s correlation coefficient. Note that these individual averages fall well within the to be expected range given that the neural tracking estimates are based on relatively short sentences (i.e., duration of ~2.5 sec) (O’Sullivan et al., 2014).

On the other hand, neural tracking of attended speech showed a moderately high, r = .64, T1–T2 correlation even outside of latent modelling. Note that the magnitude of this T1–T2 reliability is close to the short-term test-retest reliability recently reported by Panela et al. (2023). Still, when including neural tracking of attended speech in the bivariate model of change, the change-change correlation with response speed was now estimated as close to 0 (φ = –.03, n.s). This observation suggests that manifest-level high re-test reliability does not necessarily improve chances of observing a significant change-change correlation.

Lastly, we would like to point out that these bivariate model results also help to shed light on the question of whether non-linear effects of age on neural / behavioural change may affect the chance of observing a systematic change-change relationship. As shown in Fig. 6C, for neural tracking of attended speech, we observed a fairly consistent longitudinal increase across age groups. Yet, as detailed above, the change-change correlation was virtually absent.

In sum, these new results provide compelling evidence for the absence of a systematic changechange relationship.

The respective control analysis results section reads as follows, and is accompanied by Figure 6 reproduced below:

"Control analyses: The weak correlation of behavioural and neural change is robust against different quantifications of neural filtering

Taken together, our main analyses revealed that inter-individual differences in behavioural change could only be predicted by baseline age and baseline behavioural functioning, and did not correlate with contemporaneous neural changes.

However, one could ask in how far core methodological decisions taken in the current study, namely our focus on (i) the differential neural tracking of relevant vs. irrelevant speech as proxy of neural filtering, and (ii) on its trait-level characterization that averaged across different spatial-attention conditions may have impacted these results. Specifically, if the neural filtering index (compared to the neural tracking of attended speech alone) is found to be less stable generally, would this also impact the chances of observing a systematic change-change relationship? Relatedly, did the analysis of neural filtering across all trials underestimate the effects of interest?

To evaluate the impact of these consideration on our main findings, we conducted two additional control analyses: First, we repeated the main analyses using the neural filtering index (and response speed) averaged across selective-attention trials, only. Second, we repeated the main analyses using the neural tracking of attended speech, again averaged across selective-attention trials, only.

As shown in Figure 6, taken together, the control analyses provide compelling empirical support for the robustness of our main results: Linking response speed and neural filtering under selective attention strengthened their relationship at T1 (φ = .54, SE = .15, Dc2(df = 1) = 2.74, p = .1; see. Fig 6B) but did not yield any significant effects for the influence of T1 neural filtering on behavioural change (β = .13, SE = .21, Dc2(df = 1) = .43, p = .51), or for the relationship of neural and behavioural change (φ = .26, SE = .14, Dc2(df = 1) = 3.1, p = .08; please note the close correspondence to path estimates reported in Fig. 5).The second control analysis revealed a substantially higher manifest-level test-retest reliability of neural tracking of attended speech (r = .65, p<.001; Fig. 6C) compared to that of the neural tracking index. However, when linked to longitudinal changes in response speed, this analysis provided even less evidence for systematic change-related relationships: Baseline-levels of attended-speech tracking did not predict future change in response speed (β = .18, SE = .11, Dc2(df = 1) = 2.73, p = .10), and changes in neural and behavioural functioning occurred independently of one another (φ = –.03, SE = .12, Dc2(df = 1) = .06, p = .81).

In sum, the two control analyses provide additional empirical support for the results revealed by our main analysis."

**Author response image 2. sa3fig2:** Control analyses corroborate the independence of neural and behavioural trajectories under selective attention. Cross-sectional and longitudinal change in neural filtering (**A**) and neural tracking of attended speech (**C**) averaged across selective-attention trials, only. Coloured vectors (colour-coding four age groups for illustrative purposes, only) in the left subpanels show individual T1–T2 change along with the cross-sectional trend plus 95% confidence interval (CI) separately for T1 (dark grey) and T2 (light grey). Top right, correlation of T1 and T2 as measure of test-retest reliability along with the 45° line (grey) and individual data points (black circles). Bottom right, mean longitudinal change per age group and grand mean change (grey). (**B, D**) Latent change score model (LCSM) relating two-year changes in neural filtering (**B**) /neural tracking (**D**) strength to changes in response speed. Black arrows show the paths or covariances of interest that were freely estimates, grey arrows show paths that were freely estimated or fixed as part of the structural model but did not relate to the main research questions. Solid arrows indicate statistically significant effects, dashed arrows reflect nonsignificant paths. All estimates are standardised. ***p<.001, **p<.01, *p<.05.

1. The authors conclude that the temporal instability of the neural filtering measure precludes its use for diagnostic/therapeutic intervention. I agree that test-retest reliability is needed for a clinical intervention. However, given the relationship with behavior at a specific point in time, would it not be a possible target for intervention to improve performance? Even if there are different trajectories, an individual may benefit from enhanced behavioral performance in the present.

We thank the reviewer for this comment. We would agree that the observation of robust betweensubject (or even more desirable: within-subject) brain–behaviour relationships is a key desideratum in identifying potential interventional targets. At the same time, we would argue that the most direct way of evaluating a neural signature’s translational potential is by focusing on how it predicts or is linked to individual change. In revising both the Introduction and Discussion section, we hope to now better motivate our reasoning.

Other minor comments:1. Lines 106-107 What is the basis for the prediction regarding neural filtering?

In our previous analysis of T1 data (Tune et al., 2021), we found inter-individual differences in neural filtering itself, and also in its link to behaviour, to be independent of chronological age and hearing loss. On the basis of these results, we did not expect any systematic decrease or increase in neural filtering over time.

We rephrased the respective sentence as follows:

Since we previously observed inter-individual differences in neural filtering to be independent of age and hearing status, we did not expect any systematic longitudinal change in neural filtering.

1. Line 414: Replace "relevant" with "relevance".

Thank you, this has been corrected.

1. What was the range of presentation levels? Stimuli presented at 50 dB above individual sensation level could result in uncomfortably loud levels for people with mild to moderate hearing loss.

Unfortunately, we didn’t have the means to estimate the precise dB SPL level at which our stimuli were presented. Due to the use of in-ear headphones, we did not aim to measure the exact sound pressure level of presentation but instead ensured that even if stimuli were presented at the maximally possible intensity given our hardware, this would not result in subjectively uncomfortably loud stimulus presentation levels. The described procedure estimated per individual how far the maximal sound pressure level needed to be attenuated to arrive at a comfortable and easy-tounderstand presentation level.

**Reviewer #2 (Public Review):**
Summary:This study examined the longitudinal brain-behaviour link between attentional neural filtering and listening behaviour among a sample of aging individuals. The results based on the latent change score modeling showed that neither attentional neural filtering at T1 nor its T1-T2 change predicted individual two-year listening performance change. The findings suggest that neural filtering and listening behaviour may follow independent developmental trajectories. This study focuses on an interesting topic and has the potential to contribute a better understanding of the neurobiological mechanisms of successful communication across the lifespan.Strengths:Although research suggests that speech comprehension is neurally supported by an attentionguided filter mechanism, the evidence of their causal association is limited. This study addresses this gap by testing the longitudinal stability of neural filtering as a neural mechanism upholding listening performance, potentially shedding light on translational efforts aiming at the preservation of speech comprehension abilities among aging individuals.The latent change score modeling approach is appropriately used as a tool to examine key developmental questions and distinguish the complex processes underlying lifespan development in brain and behaviour with longitudinal data.Weaknesses:Although the paper does have strengths in principle, the weaknesses of the paper are that the findings are merely based on a single listening task. Since both neural and behavioral indicators are derived from the same task, the results may be applicable only to this specific task, and it is difficult to extrapolate them to cognitive and listening abilities measured by the other tasks. Therefore, more listening tasks are required to comprehensively measure speech comprehension and neural markers.The age span of the sample is relatively large. Although no longitudinal change from T1 to T2 was found at the group-level, from the cross-sectional and longitudinal change results (see Figure 3), individuals of different age groups showed different development patterns. Particularly, individuals over the age of 70 show a clear downward trend in both neural filtering index and accuracy. Therefore, different results may be found based on different age groups, especially older groups. However, due to sample limitations, this study was unable to examine whether age has a moderating effect on this brain-behaviour link.In the Dichotic listening task, valid and invalid cues were manipulated. According to the task description, the former could invoke selective attention, whereas the latter could invoke divided attention. It is possible that under the two conditions, the neural filtering index may reflect different underlying cognitive processes, and thus may differ in its predictive effect on behavioral performance. The author could perform a more in-depth data analysis on indicators under different conditions.

We thank the reviewer for their critical yet positive assessment of our work that also appreciates its potential to further our understanding of key determinants of successful communication in healthy aging. Please also see our more in-depth responses to the detailed recommendations that relate to the three main concern raised above.

Regarding the first concern of the reviewer about the limited generalizability of our brain–behaviour results, we would argue that there are two sides to this argument.

On the one hand, the results do not directly speak to the generalizability of the observed complex brain–behaviour relationships to other listening tasks. This may be perceived as a weakness. Unfortunately, as part of our large-scale projects, we did not collect data from another listening task suitable for such a generalization test. Using any additional cognitive tests would shift the focus away from the goal of understanding the determinants of successful communication, and rather speak more generally to the relationship of neural and cognitive change.

On the other hand, we would argue the opposite, namely that the focus on the same listening task is in fact a major strength of the present study: The key research questions were motivated by our timepoint 1 findings of a brain-behaviour link both at the within-subject (state) and at the between subject (trait) level (Tune et al., 2021). Notably, in the current study, we show that both, the state- and the trait-level results, were replicated at timepoint 2. This observed stability of results provides compelling empirical evidence for the functional relevance of neural filtering to the listening outcome and critically sets the stage for the inquiry into the complex longitudinal change relationships. We now spell this out more clearly in the Introduction and the Discussion.

Here, we briefly summarise how we have addressed the two remaining main concerns.

(1) Please refer to our response R1’s comment #1 on the influence of (differential) age effects on brain and behaviour. These effects were in fact already accounted for by our modelling strategy which included the continuously (rather than binned by age group) modelled effect of age. We now communicate this more clearly in the revised manuscript.

(2) We added two control analyses, one of which replicated the main analysis using selective attention trials, only. Critically, as shown in Figure 6, while the strength of the relationship of neural filtering and behaviour at a given timepoint increased, the key change-related relationships of interest remained not only qualitatively unchanged, but resulted in highly similar quantitative estimates.

**Reviewer #2 (Recommendations For The Authors):**
1. Theoretically, the relationship between brain and behavior may not be just one-way, but probably bi-directional. In this study, the authors only considered the unidirectional predictive effect of neural filtering on changes in listening task performance. However, it is possible that lower listening ability may limit information processing in older adults, which may lead to a decline in neural filtering abilities. The authors may also consider this theoretical hypothesis.

We thank the reviewer for this comment. While we did not have any specific hypotheses about influence of the behavioural state at timepoint 1 on the change in neural filtering, we ran control analysis that freely estimates the respective path (rather than implicitly assuming it to be 0). However, the results did not provide evidence for such a relationship. We report the results on p. 14 of the revised manuscript:

"We did not have any a priori hypotheses on the influence of T1 speed on the individual T1–T2 change in neural filtering. Still in a control analysis that freely estimated the respective path, we found that an individual’s latent T1 level of response speed was not predictive of the ensuing latent T1–T2 change in neural filtering (β = –.11, SE = .21, Dc2(df = 1) = .31, p = .58)."

1. The necessity of exploring the longitudinal relationship between attentional neural filtering and listening behaviour needs to be further clarified. That is, why choose attentional filtering (instead of the others) as an indicator to predict listening performance?

We are not quite certain we understood which ‘other’ metrics the reviewer was referring to here exactly. But we would like to reiterate our argument from above: we believe that focusing on neural and behavioural metrics that are (i) derived from the same task, and (ii) were previously shown to be linked at both the trait- and state-level provided strong empirical ground for our inquiries into their longitudinal change-related relationships.

Please note that we agree that the neural filtering index as a measure of attention-guided neural encoding of relevant vs. irrelevant speech signals is only one potential candidate neural measure but one that was clearly motivated by previous results. Nevertheless, in the revised manuscript we now also report on the relationship of neural tracking of attended speech and listening performance (see also our response to the reviewer’s comment #5 below).

Apart of this, by making the entire T1–T2 dataset openly available, we invite researchers to conduct any potential follow-up analyses focused on metrics not reported here.

1. Regarding the Dichotic listening task, further clarification is needed.(1) The task procedure and key parameters need to be supplemented.

We have added a new supplemental Figure S6 which details the experimental design and procedure. We have also added further listening task details to the Methods section on p.23:

At each timepoint, participants performed a previously established dichotic listening task20. We provide full details on trial structure, stimulus construction, recording and presentation in our previously published study on the first (N = 155) wave of data collection (but see also Fig. S6)12.

In short, in each of 240 trials, participants listened to two competing, dichotically presented five-word sentences spoken by the same female speaker. They were probed on the sentence-final noun in one of the two sentences. Participants were instructed to respond within a given 4 s time window beginning with the onset of a probe screen showing four alternatives. They were not explicitly instructed to respond as quickly as possible. The probe screen showed four alternative words presented either on the left or right side of the screen, indicating the probed ear. Two visual cues preceded auditory presentation (…)

We also note that the task and key parameters have been published additionally in Tune et al., 2021 and Alavash et al. (2019). We have made sure these citations are placed prominently at the beginning of the methods section.

**Author response image 3. sa3fig3:** Experimental design and procedure.

(2) Prior to the task, were the participants instructed to respond quickly and correctly? Was there a speed-accuracy trade-off? Was it possible to consider an integrated ACC-RT indicator?

We instructed participants to respond within a 4-sec time window following the response screen onset but we did not explicitly instruct them to respond as quickly as possible. We also state this more explicitly in the revised Method section on p. 23 (see also our response to comment #3 by R3 on p. 15 below).

In a between-subjects analysis we observed, both within T1 and T2, a significant positive correlation (rT1 = .33, p<.01; rT2 = .40, p<.001) of participants’ overall accuracy and response speed, speaking against a speed-accuracy trade-off. For this reason, we did not consider an integrated speed–accuracy measure as behavioural indicator for modelling.

(3) The correlation between neural filtering at T1 and T2 was weak, which may be due to the low reliability of this indicator. The generally low reliability of the difference score is a notorious measurement problem recognized in the academic community.

We fully agree with the reviewer on their assessment of notoriously noisy difference scores. It is the very reason that motivated our application of the latent change score model approach. This framework elegantly supersedes the manual calculation of differences scores, and by explicitly

modelling measurement error also removes the impact of varying degrees of reliability on the estimation of change and how it varies as a function of different influences.

While we had already detailed this rationale in the original manuscript, we now more prominently describe the advantages of the latent variable approach in the first paragraph of the Results section:

Third and final, we integrate and extend the first two analysis perspectives in a joint latent change score model (LCSM) to most directly probe the role of neural filtering ability as a predictor of future attentive listening ability. Addressing our key change-related research questions at the latent rather than the manifest level supersedes the manual calculation of notoriously noisy differences scores, and effectively removes the influence of each metric’s reliability on the estimation of change-related relationships.

We also kindly refer the reviewer to our in-depth response to R1’s comment #2 regarding the concern of neural filtering’s low test-rest reliability and its impact on estimating change-change relationships.

1. For the latent change score model, it is recommended that the authors:

This information has been added to Figure 5.

(2) In Figure 5 and Figure S2, why should the two means of the observed 2nd half scores be estimated?

In longitudinal modelling, special care needs to be applied to the pre-processing/transformation of raw data for the purpose of change score modelling. While it is generally desirable to bring all variables onto the same scale (typically achieved by standardising all variables), one needs to be careful not to remove the mean differences of interest in such a data transformation step. We therefore followed the procedure recommended by Little (2013) and rescaled variables stacked across T1 and T2 using the proportion of maximum scale (‘POMS’) methods. This procedure, however, results in mean values per timepoint ≠ 0, so the mean of the second half needed to be freely estimated to avoid model misfit. Note that the mean of the first half manifest variables was set to 0 (using the ‘marker method’; see Little, 2013) to ensure model identification.

We have added the following more detailed description to the Method section on p. 26:

To bring all manifest variables onto the same scale while preserving mean differences over time, we first stacked them across timepoint and then rescaled them using the proportion of maximum scale (‘POMS’) method99,100(…)Given our choice of POMS-transformation of raw to preserve mean differences over time, the mean of the second manifest variable had to be freely estimated (rather than implicitly assumed to be 0) to avoid severe model misfit.

(3) The authors need to clarify whether the latent change factor in Figure 5 is Δ(T1-T2) or Δ(T2-T1)?

Thank you for this comment. Our notation here was indeed confusing. The latent change factor quantifies the change from T1 to T2, so it is Δ(T2–T1). We have accordingly re-named the respective latent variables in all corresponding figures.

1. For data analysis, the author combined the trials under different conditions (valid and invalid cues) in the dichotic listening task and analyzed them together, which may mask the variations between different attention levels (selective vs. divided attention). It is recommended that the authors analyze the relationship between various indicators under different conditions.

We thank the reviewer for this comment which prompted us to (i) more clearly motivate our decision to model neural filtering across all trials, and (ii) nevertheless report the results of an additional control analyses that focused on neural filtering (or the neural tracking of attended speech) in selective-attention trials, only.

Our decision to analyse neural filtering across all spatial-attention conditions was motivated by two key considerations: First, previous T1 results (Tune et al., 2021) suggested that irrespective of the spatial-attention condition, stronger neural filtering boosted behavioural performance. Second, analysing neural filtering (and associated behaviour) across all trials provided the most direct way of probing the trait-like nature of individual neural filtering ability.

We have included the following paragraph to the Results section on p. 6 to motivate this decision more clearly:

Our main analyses focus on neural filtering and listening performance averaged across all trials and thereby also across two separate spatial-attention conditions. This choice allowed us to most directly probe the trait-like nature and relationships of neural filtering. It was additionally supported by our previous observation of a general boost in behavioural performance with stronger neural filtering, irrespective of spatial attention.

On the other hand, one could argue that the effects of interest are underestimated by jointly analysing neural and behavioural functioning derived from both selective- and divided-attention conditions. After all, it is reasonable to expect a more pronounced neural filtering response in selective-attention trials.

For this reason, we now report, in the revised version, two additional control analyses that replicate the key analyses for the neural filtering index and for the tracking of attended speech, both averaged across selective-attention trials, only:In summary, analysing neural filtering under selective attention strengthened the brain-behaviour link within a given time-point but resulted in highly similar quantitative estimated for the key relationships of interest. The analysis of attended speech tracking notably improved the neural metric’s manifest-level re-test reliability (r = .64, p<.001) – but resulted in an estimated change-change correlation close to 0.

Taken together, these control analyses provide compelling support for our main conclusion that neural and behavioural functioning follow largely independent developmental trajectories.

We kindly refer the reviewer to our detailed response to R1 for the text of the added control analysis section on p. 4f. above. The additional Figure 6 is reproduced again below for the reviewer’s convenience.

**Author response image 4. sa3fig4:** Control analyses corroborate the independence of neural and behavioural trajectories under selective attention. Cross-sectional and longitudinal change in neural filtering (**A**) and neural tracking of attended speech (**C**) averaged across selective-attention trials, only. Coloured vectors (colour-coding four age groups for illustrative purposes, only) in the left subpanels show individual T1–T2 change along with the cross-sectional trend plus 95% confidence interval (CI) separately for T1 (dark grey) and T2 (light grey). Top right, correlation of T1 and T2 as measure of test-retest reliability along with the 45° line (grey) and individual data points (black circles). Bottom right, mean longitudinal change per age group and grand mean change (grey). (**B, D**) Latent change score model (LCSM) relating two-year changes in neural filtering (**B**) /neural tracking (**D**) strength to changes in response speed. Black arrows show the paths or covariances of interest that were freely estimates, grey arrows show paths that were freely estimated or fixed as part of the structural model but did not relate to the main research questions. Solid arrows indicate statistically significant effects, dashed arrows reflect nonsignificant paths. All estimates are standardised. ***p<.001, **p<.01, *p<.05.

Figure 6 has also been supplemented by two additional figures showing behavioural functioning (Fig. S4) and neural tracking of ignored speech (Fig. S5) under selective-attention trials, only. These figures are reproduced below for the reviewer’s convenience.

**Author response image 5. sa3fig5:** Cross-sectional and longitudinal change in listening behaviour under selective attention.

**Author response image 6. sa3fig6:** Cross-sectional and longitudinal change in neural tracking of ignored speech under selective attention.

1. As can be seen from the Methods section, there were still other cognitive tasks in this database that can be included in the data analysis to further determine the predictive validity of neural filtering.

We kindly refer the reviewer to our response to their public review and comment # 2 above where we motivate our decision to focus on manifest indicators of neural and behavioural functioning that are derived from the same task.

We believe that the analysis of several additional indicators of cognitive functioning would have distracted from our main goal of the current study focused on understanding how individual trajectories of listening performance may be explained and predicted.

1. "Magnitudes > 1 are taken as moderate, > 2.3 as strong evidence for either of the alternative or null hypotheses, respectively." Which papers are referenced by these criteria? The interpretation of BF values seems inconsistent with existing literature.

It may deserve emphasis that these are log Bayes Factors (logBF). Our interpretation of logarithmic Bayes Factors (logBF) follows Lee and Wagenmakers’ (2013) classic heuristic scheme for the interpretation of (non-logarithmic, ‘raw’) BF10 values. We have added the respective reference to the manuscript.

**Reviewer #3 (Public Review):**
Summary:The study investigates the longitudinal changes in hearing threshold, speech recognition behavior, and speech neural responses in 2 years, and how these changes correlate with each other. A slight change in the hearing threshold is observed in 2 years (1.2 dB on average) but the speech recognition performance remains stable. The main conclusion is that there is no significant correlation between longitudinal changes in neural and behavioral measures.Strengths:The sample size (N>100) is remarkable, especially for longitudinal studies.Weaknesses:The participants are only tracked for 2 years and relatively weak longitudinal changes are observed, limiting how the data may shed light on the relationships between basic auditory function, speech recognition behavior, and speech neural responses.SuggestionsFirst, it's not surprising that a 1.2 dB change in hearing threshold does not affect speech recognition, especially for the dichotic listening task and when speech is always presented 50 dB above the hearing threshold. For the same listener, if the speech level is adjusted for 1.2 dB or much more, the performance will not be influenced during the dichotic listening task. Therefore, it is important to mention in the abstract that "sensory acuity" is measured using the hearing threshold and the change in hearing threshold is only 1.2 dB.

We thank the reviewer for this comment. We have added the respective information to the abstract and have toned down our interpretation of the observed behavioural stability despite the expected decline in auditory acuity.

Second, the lack of correlation between age-related changes in "neuronal filtering" and behavior may not suggest that they follow independent development trajectories. The index for "neuronal filtering" does not seem to be stable and the correlation between the two tests is only R = 0.21. This low correlation probably indicates low test-retest reliability, instead of a dramatic change in the brain between the two tests. In other words, if the "neuronal filtering" index only very weakly correlates with itself between the two tests, it is not surprising that it does not correlate with other measures in a different test. If the "neuronal filtering" index is measured on two consecutive days and the index remains highly stable, I'm more convinced that it is a reliable measure that just changes a lot within 2 years, and the change is dissociated with the changes in behavior.The authors attempted to solve the problem in the section entitled "Neural filtering reliably supports listening performance independent of age and hearing status", but I didn't follow the logic. As far as I could tell, the section pooled together the measurements from two tests and did not address the test-retest stability issue.

Please see our detailed response to R1’s comment #2 regarding the concern of how low (manifestlevel) reliability of our neural metric may have impacted the chance of observing a significant changechange correlation.

In addition, we would like to emphasize that the goal of the second step of our analysis procedure, featuring causal mediation analysis, was not to salvage the perhaps surprisingly low reliability of neural filtering. Instead, this section addressed a different research question, namely, whether the link of neural filtering to behaviour would hold across time, irrespective of the observed stability of the measure itself. The stability of the observed between-subjects brain-behaviour relationships was assessed by testing for an interaction with timepoint.

We have revised the respective Results section to more clearly state our scientific questions, and how our analysis procedure helped to address them:

"The temporal instability of neural filtering challenges its status as a potential trait-like neural marker of attentive listening ability. At the same time, irrespective of the degree of reliability of neural filtering itself, across individuals it may still be reliably linked to the behavioural outcome (see Fig. 1). This is being addressed next.

On the basis of the full T1–T2 dataset, we aimed to replicate our key T1 results and test whether the previously observed between-subjects brain-behaviour relationship would hold across time: We expected an individual’s neural filtering ability to impact their listening outcome (accuracy and response speed) independently of age or hearing status12. (…) To formally test the stability of direct and indirect relationships across time, we used a moderated mediation analysis. In this analysis, the inclusion of interactions by timepoint tested whether the influence of age, sensory acuity, and neural filtering on behaviour varied significantly across time."

Third, the behavioral measure that is not correlated with "neuronal filtering" is the response speed. I wonder if the participants are asked to respond as soon as possible (not mentioned in the method).If not, the response speed may strongly reflect general cognitive function or a personal style, which is not correlated with the changes in auditory functions. This can also explain why the hearing threshold affects speech recognition accuracy but not the response speed (lines 263-264).

Participants were asked to response within a given time window limited to 4 s but were not implicitly instructed to respond as quickly as possible. This is now stated more clearly in the Methods section (please also refer to our response to R2 on a similar question). It is important to emphasize—as shown in Figure 4A and Figure 5B —both at the manifest and latent variable level neural filtering (and in fact also the neural tracking of attended speech, see Fig. 6C) was reliably linked to response speed at T1 and T2. These results providing important empirical ground for the question of whether changes in neural filtering are systematically related to changes in response speed, and whether the fidelity of neural filtering at T1 represents a precursor of behavioural changes.

Moreover, an interpretation of response speed as an indicator of general cognitive function is not at all incompatible with the cognitive demands imposed by the task. As the reviewer rightly stated above, performance in a dichotic listening task does not simply hinge on how auditory acuity may limit perceptual encoding of speech inputs but also on how the goal-directed application of attention modulates the encoding of relevant vs. irrelevant inputs. We here focus on one candidate neural strategy we here termed ‘neural filtering’ in line with an influential metaphor of how auditory attention may be neurally implemented (Cherry, 1953; Erb & Obleser, 2020; Fernandez-Duque & Johnson, 1999).

**Reviewer #3 (Recommendations For The Authors):**
Other issues:The authors should consider using terminology that the readers are more familiar with and avoid unsubstantiated claims.For example, the Introduction mentions that "The observation of such brain-behaviour relationships critically advances our understanding of the neurobiological foundation of cognitive functioning. Their translational potential as neural markers predictive of behaviour, however, is often only implicitly assumed but seldomly put to the test. Using auditory cognition as a model system, we here overcome this limitation by testing directly the hitherto unknown longitudinal stability of neural filtering as a neural compensatory mechanism upholding communication success."For the first sentence, please be clear about which aspects of "our understanding of the neurobiological foundation of cognitive functioning" is critically advanced by such brain-behaviour relationships, and why such brain-behaviour relationships are so critical given that so many studies have analyzed brain-behaviour relationships. The following two sentences seem to suggest that the current study is a translational study, but the later questions do not seem to be quite translational.

The uncovering of robust between- and within-subject brain behaviour-relationships is a key scientific goal that unites basic and applied neuroscience. From a basic neuroscience standpoint, the observation of such brain–behaviour links provides important mechanistic insight into the neurobiological implementation of higher order cognition – here the application of auditory spatial attention in the service of speech comprehension. At the same time, they provide fruitful ground for translational inquiries of applied neuroscience. We therefore don’t consider it contradictory at all that the current study addressed both more basic and applied/translational neuroscientific research questions.

We have rephrased the respective section as follows:

"The observation of such brain–behaviour relationships critically advances our understanding of the neurobiological foundation of cognitive functioning by showing, for example, how neural implementations of auditory selective attention support attentive listening. They also provide fruitful ground for scientific inquiries into the translational potential of neural markers. However, the potency of neural markers to predict future behavioural outcomes is often only implicitly assumed but seldomly put to the test15."

More importantly, "neuronal filtering" is a key concept in the paper but I'm not sure what it means. The authors have only mentioned that auditory cognition is a model system for "neuronal filtering", but not what "neuronal filtering" is. Even for auditory cognition, I'm not sure what "neuronal filtering" is and why the envelope response is representative of "neuronal filtering".

As spelled out in the Introduction, we define our ‘neural filtering’ metric of interest as neural manifestation of the attention-guided segregation of behaviourally relevant from irrelevant sounds. By terming this signature neural ‘filtering’, we take up on a highly influential algorithmic metaphor of how auditory attention may be implemented at the neurobiological level (Cherry, 1953; Erb & Obleser, 2020; Fernandez-Duque & Johnson, 1999).

We now provide more mechanistic detail in our description of the neural filtering signature analysed in the current study:

"Recent research has focused on the neurobiological mechanisms that promote successful speech comprehension by implementing ‘neural filters’ that segregate behaviourally relevant from irrelevant sounds. Such neural filter mechanisms act by selectively increasing the sensory gain for behaviourally relevant inputs or by inhibiting the processing of irrelevant inputs5-7. A growing body of evidence suggests that speech comprehension is neurally supported by an attention-guided filter mechanism that modulates sensory gain and arises from primary auditory and perisylvian brain regions: By synchronizing its neural activity with the temporal structure of the speech signal of interest, the brain ‘tracks’ and thereby better encodes behaviourally relevant auditory inputs to enable attentive listening 8-11."

Figure 1C should be better organized and the questions mentioned in the Introduction should be numbered.

We have revised both the respective section of the Introduction and corresponding Figure 1 in line with the reviewer’s suggestions. The revised text and figure are reproduced below for the reviewer’s convenience:

"First, by focusing on each domain individually, we ask how sensory, neural, and behavioural functioning evolve cross-sectionally across the middle and older adult life span (Fig. 1B). More importantly, we also ask how they change longitudinally across the studied two-year period (Fig. 1C, Q1), and whether aging individuals differ significantly in their degree of change (Q2). We expect individuals’ hearing acuity and behaviour to decrease from T1 to T2. Since we previously observed inter-individual differences in neural filtering to be independent of age and hearing status, we did not expect any systematic longitudinal change in neural filtering.

Second, we test the longitudinal stability of the previously observed age- and hearing-loss–independent effect of neural filtering on both accuracy and response speed (Fig. 1A). To this end, we analyse the multivariate direct and indirect relationships of hearing acuity, neural filtering and listening behaviour within and across timepoints.

Third, leveraging the strengths of latent change score modelling16,17, we fuse cross-sectional and longitudinal perspectives to probe the role of neural filtering as a precursor of behavioural change in two different ways: we ask whether an individual’s T1 neural filtering strength can predict the observed behavioural longitudinal change (Q3), and whether two-year change in neural filtering can explain concurrent change in listening behaviour (Q4). Here, irrespective of the observed magnitude and direction of T1–T2 developments, two scenarios are conceivable: Intra-individual neural and behavioural change may be either be correlated—lending support to a compensatory role of neural filtering—or instead follow independent trajectories18 (see Fig. 1C)."

**Author response image 7. sa3fig7:** Schematic illustration of key assumptions and research questions. (**A**) Listening behaviour at a given timepoint is shaped by an individuals’ sensory and neural functioning. Increased age decreases listening behaviour both directly, and indirectly via age-related hearing loss. Listening behaviour is supported by better neural filtering ability, independently of age and hearing acuity. (**B**) Conceptual depiction of individual two-year changes along the neural (blue) and behavioural (red) domain. Thin coloured lines show individual trajectories across the adult lifespan, thick lines and black arrows highlight two-year changes in a single individual. (**C**) Left, Schematic diagram highlighting the key research questions detailed in the introduction and how they are addressed in the current study using latent change score modelling. Right, across individuals, co-occurring changes in the neural and behavioural domain may be correlated (top) or independent of one another (bottom).

Figure 3, the R-value should also be labeled on the four main plots.

This information has been added to Figure 3, reproduced below.

**Author response image 8. sa3fig8:** Characterizing cross-sectional and longitudinal change along the auditory sensory (**A**), neural (**B**), and behavioural (**C, D**) domain. For each domain, coloured vectors (colour-coding four age groups for illustrative purposes, only) in the respective left subpanels show an individual’s change from T1 to T2 along with the cross-sectional trend plus 95% confidence interval (CI) separately for T1 (dark grey) and T2 (light grey). Top right subpanels: correlation of T1 and T2 as measure of test-retest reliability along with the 45° line (grey) and individual data points (black circles). Bottom right panels: Mean longitudinal change per age group (coloured vectors) and grand mean change (grey). Note that accuracy is expressed here as proportion correct for illustrative purposes, but was analysed logit-transformed or by applying generalized linear models.

T1 and T2 should be briefly defined in the abstract or where they first appear.

We have changed the abstract accordingly.

References

Alavash, M., Tune, S., & Obleser, J. (2019). Modular reconfiguration of an auditory control brain network supports adaptive listening behavior. [Clinical Trial]. Proceedings of the National Academy of Science of the United States of America, 116(2), 660-669. https://doi.org/10.1073/pnas.1815321116

Cherry, E. C. (1953). Some experiments on the recognition of speech, with one and with two ears. The Journal of the Acoustical Society of America, 25(5), 975-979. https://doi.org/10.1121/1.1907229

Erb, J., & Obleser, J. (2020). Neural filters for challening listening situations. In M. Gazzaniga, G. R. Mangun, & D. Poeppel (Eds.), The cognitive neurosciences (6th ed.). MIT Press.

Fernandez-Duque, D., & Johnson, M. L. (1999). Attention metaphors: How metaphors guide the cognitive psychology of attention. Cognitive Science, 23(1), 83-116. https://doi.org/10.1207/s15516709cog2301_4

O’Sullivan, J. A., Power, A. J., Mesgarani, N., Rajaram, S., Foxe, J. J., Shinn-Cunningham, B. G., Slaney, M., Shamma,

S. A., & Lalor, E. C. (2014). Attentional Selection in a Cocktail Party Environment Can Be Decoded from Single-Trial EEG. Cerebral Cortex, 25(7), 1697-1706. https://doi.org/10.1093/cercor/bht355

Panela, R. A., Copelli, F., & Herrmann, B. (2023). Reliability and generalizability of neural speech tracking in younger and older adults. Nature Communications, 2023.2007.2026.550679. https://doi.org/10.1101/2023.07.26.550679

Tune, S., Alavash, M., Fiedler, L., & Obleser, J. (2021). Neural attentional-filter mechanisms of listening success in middle-aged and older individuals. Nature Communications, 1-14. https://doi.org/10.1038/s41467021-24771-9